# Principled neuromorphic reservoir computing

**Denis Kleyko** ®[1,2] ✉**, Christopher J. Kymn[3], Anthony Thomas[3,4], Bruno A. Olshausen[3], Friedrich T. Sommer** ®[3,5] ✉ **& E. Paxon Frady[5]**

Reservoir computing advances the intriguing idea that a nonlinear recurrent neural circuit—the reservoir—can encode spatio-temporal input signals to enable efficient ways to perform tasks like classification or regression. However, recently the idea of a monolithic reservoir network that simultaneously buffers input signals and expands them into nonlinear features has been challenged. A representation scheme in which memory buffer and expansion into higher-order polynomial features can be configured separately has been shown to significantly outperform traditional reservoir computing in prediction of multivariate time-series. Here we propose a configurable neuromorphic representation scheme that provides competitive performance on prediction, but with significantly better scaling properties than directly materializing higher-order features as in prior work. Our approach combines the use of randomized representations from traditional reservoir computing with mathematical principles for approximating polynomial kernels via such representations. While the memory buffer can be realized with standard reservoir networks, computing higher-order features requires networks of 'Sigma-Pi' neurons, i.e., neurons that enable both summation as well as multiplication of inputs. Finally, we provide an implementation of the memory buffer and Sigma-Pi networks on Loihi 2, an existing neuromorphic hardware platform.

Reservoir computing is a paradigm for computing with recurrent neural circuits that are inspired by observations in neuroscience[1,2] and has yielded efficient realizations of recurrent neural networks, an architecture ubiquitous in technical applications for processing multivariate time-series. Reservoir computing uses a neural dynamical system, the so-called "reservoir," to map a time-series into a pattern in a high-dimensional state space, which is then fed into a one-layer neural network[3,4]. The one-layer network can be trained in a supervised fashion to perform tasks, such as classification or regression of time-series. The reservoir is thought to serve two purposes (Fig. 1a): First, it is a memory buffer for the input signals, often a fading memory buffer if emphasis on the recent input history is desired. For buffering it is

crucial that the dynamics of the reservoir are fixed. For example, the standard strategy is to use a recurrent network with fixed random connections[4,5]. Second, nonlinearities in the reservoir dynamics can enable rich feature spaces[6], including nonlinear functions of the input signals, potentially leading to separability and generalization unachievable on the original signals.

In practice, however, the ability of reservoir networks to form rich feature spaces could be limited. For example, reservoir networks with common saturating neural activation functions mainly cause memory fading, and the resulting feature space still closely resembles those of linear recurrent networks[7–9]. Reservoir networks containing more neurobiological details, such as spiking neurons[1], or synaptic connections

[1]Centre for Applied Autonomous Sensor Systems, Örebro University, Örebro, Sweden. [2]Intelligent Systems Lab, RISE Research Institutes of Sweden, Kista, Sweden. [3]Redwood Center for Theoretical Neuroscience, University of California, Berkeley, CA, USA. [4]Electrical and Computer Engineering, University of California, Davis, CA, USA. [5]Neuromorphic Computing Lab, Intel, Santa Clara, CA, USA. ✉e-mail: denis.kleyko@oru.se; fsommer@berkeley.edu

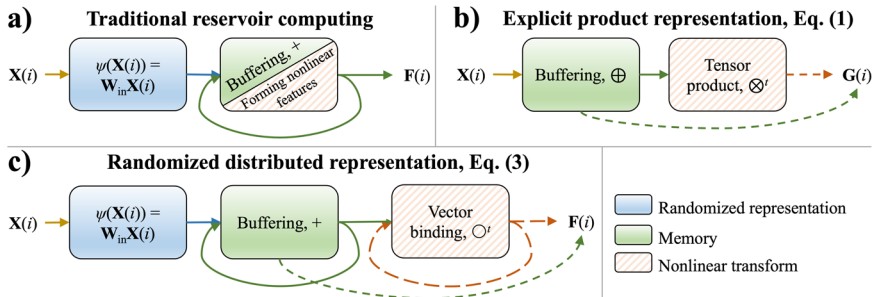

**Fig. 1 | Overview of three different representation schemes. a** Traditional reservoir networks simultaneously buffer input signals and extract nonlinear features from them. **b** The product representation[12] computes the memory buffer and the higher-order features explicitly using concatenation and tensor product operations, respectively. **c** The proposed approach can form randomized distributed representations of higher-order features using the representations of the reservoir computing network. The solid lines denote the compulsory connections while the dashed lines are the optional ones. The diagrams only show the parts of the models involved in computing the representations of the feature space.

with short-term plasticity as additional dynamic variables[10], can create richer representations. However, the richness is difficult to adjust for serving a particular computational task in the best possible way. Illustrating these limitations of traditional reservoir computing, it has been recently shown[11,12] that a representation scheme that computes tensor products between time-delayed states of the input signals (Fig. 1b) can empirically outperform traditional reservoir networks on an important task of predicting dynamical systems.

In light of these limitations, here we investigate principled, configurable, and efficient ways to implement reservoir computing with nonlinear features on neuromorphic hardware. We propose a bipartite approach combining two generic neural circuits (Fig. 1c): traditional reservoir networks for forming a memory buffer, and novel Sigma-Pi networks[13,14] for computing nonlinear features. We theoretically characterize the two essential operations for jointly representing feature spaces – concatenation and tensor product – and show that each operation results in a different similarity structure between the constructed representations. We formulate a concrete scheme based on randomized distributed representations for multivariate time-series prediction and demonstrate these networks implemented on the neuromorphic chip Loihi 2[15]. The proposed approach, which builds on ideas from vector symbolic architectures[16–18] and randomized kernel approximation[19,20], can form representations with approximately the same similarity structure as concatenation or tensor product, but the dimensionality of the representation remains fixed. We evaluate the novel randomized distributed representations on the prediction of chaotic dynamical systems and show that often the same quality of predictions can be achieved with representations that need fewer dimensions than in the original, explicit approach[12].

## Results
### Joint representations formed by concatenation or tensor product

In order to make predictions from time-series, one must create a representation that includes some history of the input signals as well as relevant, nonlinear features. A classic approach to represent the history of a trajectory is to form a memory buffer by concatenating $d$-dimensional state vectors observed at different time points[21], e.g.: $\mathbf{V}_C^{(i)} = \mathbf{X}(i-1) \oplus \mathbf{X}(i) = [X_1(i-1), \ldots, X_d(i-1), X_1(i), \ldots, X_d(i)]$. In this resulting feature space, the inner product between the representations of two trajectories formed by concatenation equals the *sum* of inner products between the state vectors at the single time points:

$$\left\langle \mathbf{V}_C^{(i)}, \mathbf{V}_C^{(j)} \right\rangle = \sum_d X_d(i-1)X_d(j-1) + \sum_d X_d(i)X_d(j) = \langle \mathbf{X}(i-1), \mathbf{X}(j-1) \rangle + \langle \mathbf{X}(i), \mathbf{X}(j) \rangle.$$

An alternative is to combine state vectors at different time points in a trajectory by the tensor product. The resulting representation contains products of the components of the original state vectors, i.e., nonlinear higher-order features, e.g.: $\mathbf{V}_T^{(i)} = \mathbf{X}(i-1)\otimes \mathbf{X}(i) = [X_1(i-1)X_1(i), X_1(i-1)X_2(i), \ldots, X_d(i-1)X_d(i)]$. The inner product between a pair of tensor products of trajectories, the Frobenius inner product, corresponds to the *product* of inner products between the observable states at each point in time:

$$\left\langle \mathbf{V}_T^{(i)}, \mathbf{V}_T^{(j)} \right\rangle = \sum_{d,d'} X_d(i-1)X_{d'}(i)X_d(j-1)X_{d'}(j)$$
$$= \langle \mathbf{X}(i-1), \mathbf{X}(j-1) \rangle \cdot \langle \mathbf{X}(i), \mathbf{X}(j) \rangle.$$

The similarity measured by the inner product is qualitatively different for the two types of representations. With concatenation of the input signals, a pair of trajectories has non-negligible similarity even if the signals coincide in just a single time point. Conversely, the tensor product forms polynomial products of the input signals[22], and as a result, the similarity between trajectories is high only if they coincide in all time points. Thus, for solving concrete computational tasks it is important to flexibly combine the two operations on the input signals into the final feature space.

### Representing trajectories with concatenation and tensor product

Any concrete representation scheme to encode multivariate time-series by combining concatenation and the tensor product is a task-specific design choice. Here we formalize such a representation scheme recently proposed for the efficient prediction of chaotic dynamical systems[12]. A state at a single time point $i$ is described by a $d$-dimensional vector $\mathbf{X}(i)$. A trajectory includes $k$ time points. The entire trajectory is represented by a $dk$-dimensional vector, a concatenation of the $k$ $d$-dimensional state vectors. To generate higher-order features, the tensor product is applied to each trajectory representation with itself. To form $t$th-order features the tensor product must be applied $t-1$ times. Finally, concatenation is applied again to combine the features of different orders into a single vector. The representation $\mathbf{G}$ for trajectories resulting from this representation scheme can be written as:

$$\mathbf{G} = \bigoplus_{t\in\mathcal{T}} \left( vec\left( \bigotimes^t \left( \bigoplus_{l=1}^k \mathbf{X}(\mathcal{M}_l) \right) \right) \right) \in \mathbb{R}^{\sum_{t\in\mathcal{T}} \binom{dk+t}{t+1}} \quad (1)$$

where $\mathcal{M}$ is a $k$-tuple containing relative indices of time points in the trajectory to be concatenated ($\bigoplus$), $\mathbf{X}(\mathcal{M}_l)$ is the $d$-dimensional state at the time point $\mathcal{M}_l$. The order of tensor product ($\bigotimes^t$) features is

controlled by set $\mathcal{T} \subset \{0, \ldots, t, \ldots, n-1\}$ where each integer $t+1 \in \mathcal{T}$ describes a desired order of polynomial features.

We will refer to Eq. (1) as a *product representation* of polynomial higher-order features, the corresponding diagram is outlined in Fig. 1b. Akin to local representations[23], vector components in **G** represent either individual components of the state vectors or a product of a subset of such components. A severe problem with the product representation is that the scaling is poor. The dimensionality of the feature space **G** grows exponentially with $n$ – the highest order of the features that are used, i.e., for $n$th-order features, the resulting dimensionality is still $\begin{pmatrix} dk+n-1 \\ n \end{pmatrix}$ even when only the unique features are considered, "Product representation of higher-order features" section.

## Randomized distributed representations of trajectories

As shown in refs. 11,12,24, the product representation scheme Eq. (1) can outperform traditional reservoir networks in the prediction from multivariate time-series. However, a significant limitation of this approach is that the dimensionality of the representation grows exponentially with the order of the polynomial, "Product representation of higher-order features" section.

Inspired by traditional reservoir computing, which often involves randomized distributed representations of the input signals[5], we propose an approach that combines the transparency and richness of the product representation with the advantages of randomized representations[19,25,26]. To realize this idea in reservoir computing, we leverage an algebraic framework, known as hyperdimensional computing or vector symbolic architectures (VSA)[16–18]. The initial step of the approach is to embed a single time point $\mathbf{X}(i) \in \mathbb{R}^d$ into a $D$-dimensional randomized representation under a map $\psi : \mathbb{R}^d \to \mathbb{R}^D$. Here, we focus on embeddings based on linear random projection[27], which are commonly used in VSA[28,29], that transforms a $d$-dimensional state vector $\mathbf{X}(i)$ into a $D$-dimensional embedding vector via:

$$\psi(\mathbf{X}(i)) = \mathbf{W}_{\text{in}}\mathbf{X}(i) \in \mathbb{R}^D, \qquad (2)$$

Here, $\mathbf{W}_{\text{in}} \in \mathbb{R}^{D \times d}$ is a random projection matrix, where each column is chosen i.i.d. from a certain distribution, depending on the choice of a VSA model as discussed below. Such a randomized linear embedding is also the standard first step in traditional reservoir computing.

The VSA framework offers three standard algebraic operations for manipulating randomized distributed representations: vector superposition (denoted as $+/\sum$), vector binding (denoted as $\circ/\bigcirc$), and permutation (denoted as $\rho$). The exact instantiation of these operations depends on the particular VSA model[30,31] and affects the corresponding implementation as discussed in "Networks of Sigma-Pi neurons for tensor product and binding" section. The advantage of VSA is that superposition, binding, and permutation are dimensionality-preserving – independent from how these operations are combined and iterated, the dimensionality of the resulting representation is always equal to $D$, the dimensionality chosen in the random projection step, Eq. (2).

Regardless of the type of VSA model, there is a correspondence between the similarity structures of features formed in the VSA space to the similarity structure of concatenation and tensor product. As described in ref. 32, superposition protected by permutation, akin to the recurrent computation in reservoir computing, has approximately the same similarity structure as concatenation. Thus, a reservoir network as well as a VSA-protected superposition are, generally, like concatenation of inputs and, in essence, form a feature space that acts like a short-term memory of the input history (cf. Fig. 1a, c). Such a short-term memory can be extended to the concept of a *fading memory buffer*, where inputs from older time points gradually fade

away. The time constant of this fading memory can be controlled by either point-wise saturating nonlinearities or by the recurrent weight matrix[8]. The memory buffer retains the similarity structure like concatenation, but one needs to account for the fading memory property.

Similarly, vector binding has a matching similar structure as the tensor product[32]. Thus, in the VSA framework, the randomized and distributed version of the product representation Eq. (1) is given by:

$$\mathbf{F} = \sum_{t \in \mathcal{T}} \left( \bigcirc^t \left( \sum_{l=1}^k \mathbf{W}_\rho^{l-1}\big(\mathbf{W}_{\text{in}}\mathbf{X}(\mathcal{M}_l)\big) \right) \right) \in \mathbb{R}^D, \qquad (3)$$

where $\mathbf{W}_\rho$ is a mixing matrix with the "echo-state property"[3] applied $l-1$ times. Practically, $\mathbf{W}_\rho$ can be a simple permutation[33,34], which we denote as $\rho(\cdot)$. The diagram of the proposed approach is depicted in Fig. 1c while further step-by-step details of this representation scheme are provided in "Randomized representations of higher-order features with binding" section. The crucial idea is that distributed representations are able to approximate the same similarity structure as the product representation above, but in a much more parsimonious way, "Kernel approximation guarantees for randomized representations" section. This holds because vector binding and permutation of vector components distribute over vector superposition (the basis for the "computing in superposition" principle[18]) and produce joint distributed representations where the inner product between such representations approximates the inner product between representations formed using the explicit feature map in Eq. (1). In general, as the dimensionality, $D$, grows larger, the fidelity of this approximation will improve. The precise rate at which this happens depends on the maximum polynomial degree and the particulars of the data in question[19]. We quantify this more precisely in "Kernel approximation guarantees for randomized representations" section, but the crucial feature is that to achieve any desired constant fidelity of approximation, the dimensionality $D$ depends only quadratically on the maximum polynomial degree, as opposed to the *exponential* dependence encountered when representing polynomial features explicitly by the product representation.

## Networks of Sigma-Pi neurons for tensor product and binding

In order to make use of the product representation or its equivalent distributed representation on neuromorphic hardware, there must be a network motif for computing concatenation and tensor product features. As already mentioned, the memory buffer for storing the trajectory of time-delayed states can be implemented with traditional linear reservoir networks, Fig. 1a. These networks are composed of conventional *Sigma* neurons, which sum up the synaptic inputs and potentially apply a point-wise nonlinear activation function.

To implement any variant of features formed by the tensor product, Eq. (1) or Eq. (3), a network motif is required for computing tensor product or vector binding. Such a network motif requires an additional type of neuron, *Pi* neurons[13,14], which compute the product of synaptic inputs. Thus, to compute the higher-order features of two input vectors, $\mathbf{X}, \mathbf{Y} \in \mathbb{R}^d$, one would have a population of $d \times d$ Pi neurons. Each Pi neuron would have two inputs, one from **X** and one from **Y**, in accordance with the tensor product, Fig. 2a. Similarly, features obtained via vector binding also require Pi neurons or a network of Sigma and Pi neurons. We highlight two network motifs for computing randomized distributed representations using the multiply-add-permute model (MAP)[35] as well as the sparse block code model (SBC)[32,36], see Fig. 2b, c. These different variations of motifs to compute higher-order features have different trade-offs in terms of the number of Sigma or Pi neurons needed, as well as the total number of synaptic connections. Based on the traits of the neuromorphic hardware, one variation might be more favorable than another.

Another advantage of the distributed representation approach in Fig. 2b, c is that higher-order features can be computed by using

the network motifs for binding recurrently (recurrent connections are not shown in the figure). Since the dimensionality of the result of binding is the same as the dimensionality of the inputs, the result can be sent backward as one of the inputs and recombined in the next iteration. This can greatly reduce the resource requirements for computing high-order features on neuromorphic hardware since the synaptic connections and neurons for the binding network can be reused.

Table 1 summarizes the resources – as defined by the number of neurons and synaptic connections – needed for Sigma-Pi networks. It also includes another VSA model – holographic reduced representations (HRR)[37] that is used in "Experiments on CPU" section (see Supplementary Material S-VI for the experiment comparing the performance of these models). For all networks, the initial memory buffer of time-delayed states is formed with a permutation matrix, which is the simplest matrix structure for creating an effective reservoir, requiring only a single synapse for each neuron[33,34]. The holographic reduced representations model realizes the binding through the circular convolution. This circuit requires $D^2$ Pi neurons, which first compute the tensor product between the input populations, and $D$ Sigma neurons aggregate the activity of Pi neurons. In the multiply-add-permute model, the binding operation is the component-wise Hadamard product, so $D$ Pi neurons are sufficient to implement it. For the sparse block code model, the binding operation is block-wise circular convolution, which requires $DL$ Pi neurons and $D$ Sigma neurons, where $L$ is the size of the block. Each of these randomized systems requires an embedding step ("Randomized representations of higher-order features with binding" section, Eq. (17)) to transform the input signals into distributed representations. Since both holographic reduced representations and multiply-add-permute use dense

representations, the random projection matrix must be of size $dD$. However, for the sparse block code model, a neuromorphic system can take advantage of the matrix's sparsity, requiring only $dD/L$ synaptic connections for the embedding. To compute higher-order features in all networks, the recurrent connections are added from the output of the binding stage back to one of the input populations, which can be done with one-to-one synaptic connections, requiring only $D$ extra synapses. With this recurrent motif, features of arbitrary order can be computed without requiring additional network resources, but using more iterations.

Importantly, the concrete VSA model should be chosen not only based on the corresponding network complexity but also based on its suitability to the targeted computing hardware. To demonstrate this point, "Experiments on neuromorphic hardware" section presents an implementation of the existing neuromorphic chip Loihi 2.

## Experiments on CPU

Following the example of traditional reservoir computing[1,3], Gauthier et al.[12] used the product representation scheme Eq. (1) combined with a ridge regression to construct predictive models for multivariate time-series and evaluated them on predicting chaotic dynamical systems. During training, a readout matrix ("Product representation of higher-order features" section, Eq. (9)), is obtained from the ridge regression solution (with regularization parameter $\alpha$). The training data include points in the feature space and the corresponding ground truth of the target function, e.g., the next state vector of the dynamical system as determined by conventional numerical integration.

Here we evaluate the described approaches to data representation in experiments on autoregressive prediction of chaotic dynamical systems; for details of the considered dynamical systems, see "Tasks and experimental configurations" section. For the Lorenz63 system, Eq. (23), Fig. 3a depicts a ground truth trajectory and training time points (blue dotted lines) for all three observable states that are overlaid with the training phase predictions for the product (red dashed lines) and distributed representations (green dash-dotted lines), respectively. Similar to the product representation, Fig. 3b, the strange attractor predicted by the distributed representation, Fig. 3c, highly resembles the true attractor in Fig. 3a, indicating the successful reconstruction of the dynamical system (see quantitative results in Supplementary Material S-VIII). Supplementary Material S-X shows that this dynamical system can be reconstructed successfully even in the presence of noise. Further, both approaches can closely follow the true Lorenz63 system in the short term for several Lyapunov times, Fig. 3b, c. For the double-scroll system, Eq. (24), the ground truth data from the training phase is presented in Fig. 3d. The predicted attractors in Fig. 3e, f also resemble the true attractor where both product and distributed representations closely follow the true dynamics for several Lyapunov times. For the third Mackey–Glass system, Eq. (25), in Fig. 3g, the feature space **G** for the product representation includes features up to third-order following[38], Eq. (31) where $D = 84$. However, a comparison of the predicted attractor, Fig. 3h, to the true one, Fig. 3g, reveals that predicting the system with this feature space is challenging, which is consistent with previous observations[38]. This exemplifies the poor scaling of product representations as extending **G** with

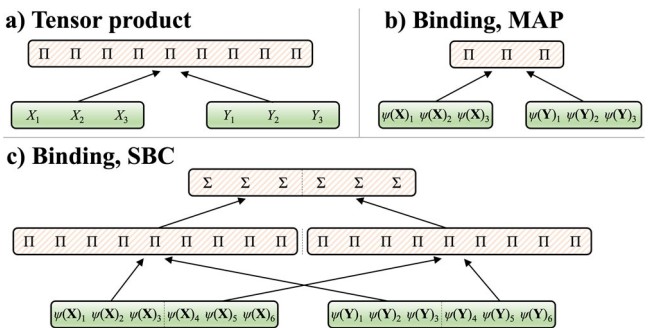

**Fig. 2 | Network motifs for computing representations of higher-order features.** The illustrated networks are shown for the concrete number of neurons in the input populations, the same principles apply to construct networks of any size. **a** Network motif for tensor product. The two $d$-dimensional inputs are combined into a $d^2$-dimensional output of Pi neurons. The panel illustrates $d = 3$. **b** Network motif for binding with the multiply-add-permute (MAP) model. The two $D$-dimensional inputs are combined into a $D$-dimensional output of Pi neurons. The panel illustrates $D = 3$. **c** Network motif for binding with the sparse block code (SBC) model. Two $D$-dimensional inputs are combined into a $D$-dimensional output, through an intermediate layer of Pi neurons. The size of the intermediate layer is $DL$, where $L$ is the size of a block. The panel illustrates $D = 6$ and $L = 3$.

## Table 1 | Resources required for computing higher-order features

| Model type: | Embedding, connections | Memory buffer, network size | Higher-order features, network size | Higher-order features, connections |
|---|---|---|---|---|
| Product representation | 0 | $dk$ | $\sum_{t \in \mathcal{T}} \binom{dk+t}{t+1}(\Pi)$ | $\mathcal{O}((dk)^t)$ |
| HRR | $dD$ | $D$ | $D(\Sigma) + D^2(\Pi)$ | $3D^2$ |
| MAP | $dD$ | $D$ | $D(\Pi)$ | $2D$ |
| SBC | $dD/L$ | $D$ | $D(\Sigma) + DL(\Pi)$ | $3DL$ |

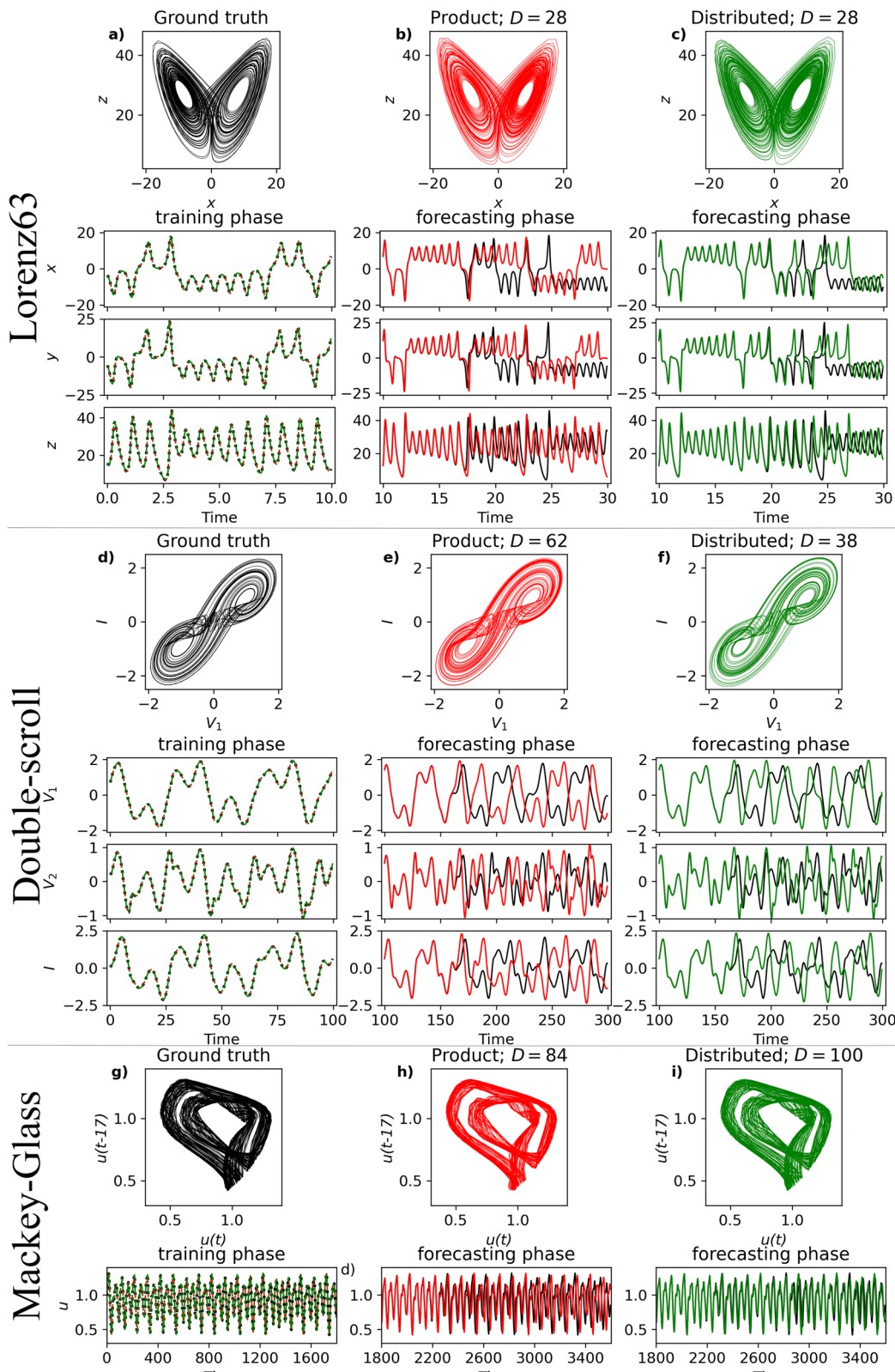

**Fig. 3 | Examples of predicting chaotic dynamical systems.** Predicting **a**–**c** Lorenz63, **d**–**f** double-scroll, and **g**–**i** Mackey–Glass dynamical systems using the product (center column) or distributed (right column) representations. True (**a**, **d**, **g**) and predicted (**b**, **c**, **e**, **f**, **h**, **i**) strange attractors. **a**, **d**, **g** Training time points (dotted lines) overlayed with the corresponding predictions for the product (dashed lines) and distributed (dash-dotted lines) representations. For the product representations, regularization parameter $\alpha$ is set to: **b** $2.5 \times 10^{-6}$, **e** $1 \times 10^{-4}$, **h** $1 \times 10^{-7}$; while for the distributed ones: **c** $1 \times 10^{-7}$, **f** $1 \times 10^{-5}$, and **i** $1 \times 10^{-6}$; see "Tasks

and experimental configurations" section for the description of the procedure for choosing the hyperparameters. Comparison of the ground truth behavior (black) to the dynamics predicted by either product (**b**, **e**, **h**) or distributed (**c**, **f**, **i**) representation. The median normalized root-mean-square error (NRMSE) across 1000 simulations over three Lyapunov times during the prediction phase for product representations is **b** $1.47 \times 10^{-2}$, **e** $1.98 \times 10^{-2}$, and **h** $3.59 \times 10^{-1}$; respectively for distributed representations: **c** $1.59 \times 10^{-2}$, **f** $2.17 \times 10^{-2}$, and **i** $1.80 \times 10^{-1}$.

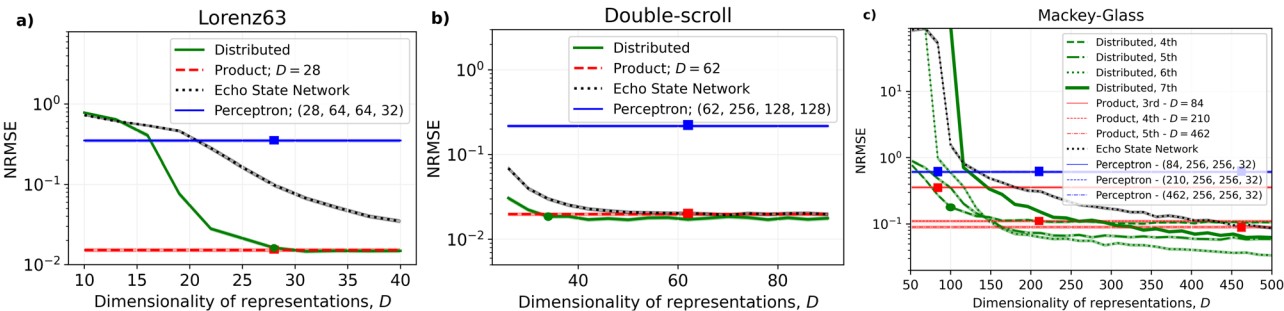

**Fig. 4 | Median predictive performance against the dimensionality of D.**
**a** Lorenz63, **b** double-scroll, and **c** Mackey–Glass systems. The configurations of the product and distributed representations depicted in Fig. 3 are marked by red squares and green circles, respectively. The median performance of the corresponding product representations from Fig. 3 is shown as thick dashed red lines where the red squares additionally emphasize the exact location of the configurations with respect to model's size that is driven by higher-order features used to form the feature space. Similarly, the performance of the multilayer perceptron is shown as a thick solid blue line with blue squares emphasizing that the size of the

first hidden layer matches that of the product representation. The echo-state network ("Experiments with traditional reservoir computing networks" section) is depicted as a dotted black line. The following hyperparameters are used: $\beta = 0.25$, $\gamma = 0.1$, $\alpha = 1 \times 10^{-7}$ (Lorenz63); $\beta = 0.1$, $\gamma = 0.1$, $\alpha = 1 \times 10^{-9}$ (double-scroll); $\beta = 0.25$, $\gamma = 1.00$, $\alpha = 1 \times 10^{-7}$ (Mackey–Glass). For each configuration, NRMSE is computed over three Lyapunov times where the reported values are obtained from 1000 randomly initialized simulations. Shaded areas show the median standard error.

fourth-order features requires $\binom{dk+3}{4} = 126$ additional features increasing the model size by 150%. This is not the case for the distributed representation, Fig. 3i, since already for $D = 100$ (19% increase to **G**) it accommodates even fourth-order features, Eq. (32), and, consequently, achieves a substantial decrease in normalized root-mean-square error (NRMSE, Eq. (35) in "Performance metrics" section) − $3.59 \times 10^{-1}$ versus $1.80 \times 10^{-1}$. Further, this dimensionality is already sufficient to start creating models that much better reconstruct the true attractor, Fig. 3i. The individual distributed representation-based models in Fig. 3 indicate that the dimensionality is an important hyperparameter affecting both model's complexity and its predictive performance measured by NRMSE. This effect is presented in Fig. 4.

For the Lorenz63 system, Fig. 4a, the product representation space includes 28 features, Eq. (27), while the number of dimensions $D$ for the distributed representation is a hyperparameter that can be set freely. In fact, for the same dimensionality ($D = 28$) the distributed representation provides performance matching the product representation baseline. This is an important result showing that randomized distributed representations attain strong performance even with low dimensions, which contradicts the existing narrative. Similar to the distributed representation, the traditional reservoir computing (echo state network, "Experiments with traditional reservoir computing networks" section) also improves with increased $D$. But in agreement with the results reported in refs. 11,12, for the Lorenz63 system it needs more dimensions to match the performance of the higher-order features (though, it is not guaranteed to be the case for every system, see Supplementary Material S-I). Furthermore, the performance of the multilayer perceptron is nowhere near the models with higher-order features neither for the Lorenz63 system nor for the other systems. For the double-scroll system, the higher-order feature spaces are constructed from first- & third-order features, Eq. (29), such that the product representation includes $D = 62$ features. The distributed representation, however, demonstrates matching performance, Fig. 4b, with just 34 dimensions resulting in 45% resource savings (see also Supplementary Material S-I for a similar result on another task). Thus, even for a moderately high number of features in **G**, the distributed representation could provide non-trivial computational savings. This advantage of the proposed approach is emphasized further for the Mackey–Glass system, Fig. 4c, where the distributed representation with 100 dimensions already accommodates fourth-order features (green circle). The predictive performance can be improved further at the cost of increased dimensionality, which emphasizes the

value of including additional features (fourth-order). In contrast to the rigid design space of the product representation (red squares), the absence of fixed dependency between the dimensionality of distributed representations and the number of features in the corresponding product representation results in a flexible design space controlling the trade-off that is demonstrated by additional models with an even higher order of features (up to seventh). As expected, the configurations of the distributed representation including more features require more dimensions to demonstrate reasonable performance. At the same time, given representations that are large enough such models demonstrate better predictive performance. Though as with the product representation, there are diminishing returns since much larger representations are required to slightly improve the performance.

## Experiments on neuromorphic hardware
Here, we demonstrate the implementation of a memory buffer and Sigma-Pi networks on the neuromorphic chip Loihi 2[15,39]. Loihi 2 (Fig. 5a) is an asynchronous neuromorphic computing architecture and communicates information with event-based packets. In Loihi 2, these packets can contain 24 bits of information – a "graded spike", which we use to transmit the magnitude of vector components. Furthermore, Loihi 2 has a programmable engine that allows users to define custom neuron models, which we use to implement both Sigma and Pi neurons.

Following "Networks of Sigma-Pi neurons for tensor product and binding" section, we utilize two basic types of neurons for the implementation: Sigma and Pi. The Sigma neuron computes the inner product between the input population and synaptic weights and transmits the result as a graded spike. The Sigma neuron can also be configured with a threshold, where the results of the inner product below the threshold will lead to no spiking output, which can reduce spike traffic. The Pi neuron is a special type of neuron that has two input channels. Synaptic inputs are accumulated on the two channels, and then the neuron computes the product of these inputs and outputs the result as a graded spike. Typically here there is only a single synaptic input on each channel.

The representations of activity in the Loihi 2 neurons are, thus, event packets containing 24 bits representing fixed-point integers. The 8-bit synaptic weights and the Pi neurons compute fixed-point multiplication by multiplying the integer values and shifting right, where the fixed point is typically $2^7$ for synaptic weights and $2^{12}$ for graded spikes. The chip is programmed using the base software package *Lava*

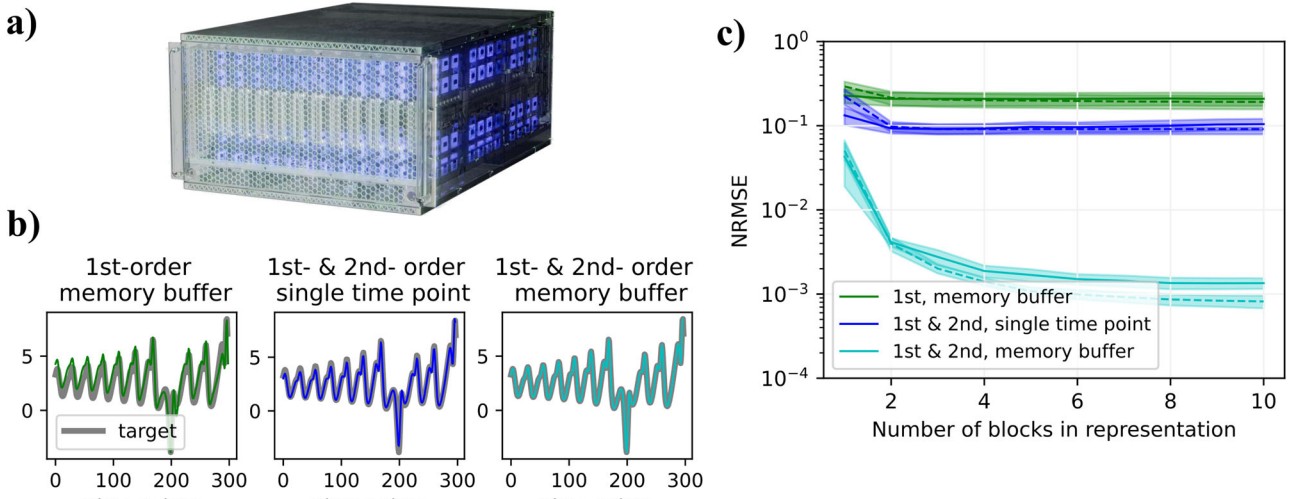

**Fig. 5 | Implementation and experiments on the neuromorphic chip Loihi 2.** **a** Hala Point packages 1152 Loihi 2 processors. The system supports up to 1.15 billion neurons and 128 billion synapses. Note that such scale far exceeds the requirements of the conducted experiments, but we wish to highlight the potential scale achievable by existing neuromorphic hardware. **b** Three network architectures based on the sparse block code model are implemented on Loihi 2 and compared. The readout of the feature space formed by the network is compared to the target function. **c** Average performance of the different network architectures is shown versus the number of blocks used in the representation ($L = 20$). Dashed lines correspond to the representations computed on the CPU. Shaded areas show the median standard error.

(https://github.com/lava-nc), as well as *Lava-VSA*, which provides tools for constructing VSA circuits. Since sparsity is an important desiderata for Loihi 2 circuits, we leverage the sparse block code model, Fig. 2c. The representations are structured into $K = 10$, $L = 20$-dimensional blocks, with $D = LK$. The *Lava-VSA* software package contains modules for creating the network motifs for computing the desired distributed representation of higher-order features that can be executed on Loihi 2.

To validate that our implementation produces meaningful representations, we perform a feasibility study by using the states of the Lorenz63 system, Eq. (23), as an input to three different network architectures running on Loihi 2. The goal is to predict a randomly chosen quadratic function that depends on two most recent Lorenz63 state vectors (Fig. 5b gray line depicts and example target function). To illustrate the proof-of-principle, we implement and compare three networks: a linear reservoir network that computes the memory buffer of recent input history (Fig. 5b, left); a network that computes the joint distributed representation of the most recent state vector and its second-order features (Fig. 5b, center); and a network that combines the memory buffer with first and second-order features (Fig. 5b, right). Figure 5b illustrates qualitatively that the first network provides the worst predictions while the predictions from the third network are nearly perfect. The same point is demonstrated quantitatively in Fig. 5c that reports the prediction error for eight random instantiations of the networks (solid lines). The performance is shown against the number of sparse blocks of fixed size ($L = 20$). As expected, neither the first nor the second network's representations are sufficient to closely predict the target function, since these feature spaces do not contain the features matching the target function. In contrast, using the representations including the memory buffer and its second-order features results in small errors that decrease with the number of blocks, as all relevant target features are present. Furthermore, the results of all three networks on Loihi 2 closely follow the results obtained on their CPU counterparts (dashed lines).

## Discussion

Reservoir computing is a powerful and general paradigm for computing with randomized distributed representations in a recurrent neural network. It draws on principles of neural computation, and it has proven useful for a wide range of tasks[40]. Yet despite general guarantees on function approximation[6], traditional reservoir computing is often difficult to interpret and optimize in practice. This motivates exploring modifications of the original architecture to achieve the same performance with less resources. For example, ref. [41] used reservoirs combined with time delays and ref. [42] used structured matrices to speed up updates of the reservoir. Another promising approach is to model higher-order polynomial features in the data explicitly. This idea is explored in refs. [11,12,24] who show that extracting higher-order features from time-series can dramatically improve the performance of reservoir computing models. While powerful, the dimensionality of this explicit formation of higher-order features grows exponentially with the order of the polynomial, making scaling to high-dimensional inputs difficult. In addition, large explicit feature spaces conflict with the classical motivation of parsimony in reservoir computing, and are less amenable for deployment on neuromorphic hardware. Here, we propose an approach that computes higher-order features implicitly and preserves the performance benefits of the explicit construction with reduced resource requirements. Furthermore, we show that this approach provides a principled way of approximating polynomial kernels with compact neural circuits and provides a proof-of-principle demonstration by implementing it on the neuromorphic chip Loihi 2.

Polynomial kernel machines and polynomial regression are widely known and useful tools in machine learning. The earlier results in refs. [11,12] and our results enrich the repertoire of reservoir computing networks, by explicitly linking reservoir networks to the polynomial kernels. The theoretical connection between Volterra series and polynomial kernel regression[43] further supports the idea of using these representations for learning dynamical systems. Our approach is principled by building on classic work from the machine learning literature on approximating kernel machines with randomized representations[19,20]. Standard kernel machines avoid the exponential cost of computing all higher-order features but still have costs quadratic in the number of data points[25] ("Implicit realization via polynomial kernel machine" section). By contrast, we use *randomized distributed* representations of polynomial kernels, which capture the

same similarity structure as explicitly forming the feature map, but in a "compressed" representation that is much more parsimonious. A crucial advantage of this approach over explicitly forming the polynomial features is that in the former, the polynomial features are stored "in superposition"[18] using fewer dimensions than would be required to explicitly represent them as in the latter. The price of this compression is that the distributed representation is approximate ("Kernel approximation guarantees for randomized representations" section): the similarity kernel recovered by the distributed representations is only a noisy version of the true kernel. The magnitude of this noise as a function of dimensionality can be quantified precisely[19,20] using the theory of concentration of measure, and the dimensionality required to achieve a small error of approximation is modest compared to explicitly representing the features. This theoretical analysis underlies our empirical findings that the proposed approach performs more accurate prediction with smaller dimensions (Fig. 4).

Prior theoretical work on randomized kernel approximation left open the question of how to best implement concatenation and tensor products on computing hardware. Our approach addresses this gap for neuromorphic hardware, leveraging vector symbolic architectures[16–18,30], an algebraic, dimensionality-preserving framework for forming compositional distributed representations. The binding and superposition operations of vector symbolic architectures correspond to approximate representations of concatenation and tensor product, respectively[32]. Our approach points out that two motifs, memory buffers and higher-order polynomial features, can be composed to form feature spaces and consolidated into two neural networks (see ref. 44 for an alternative proposal within a single network). For the second motif ("Randomized distributed representations of trajectories" section), we demonstrate that the recursive vector bindings correspond to computations of higher-order features of the time points in the memory buffer of the reservoir. In addition, we propose that a recurrently connected network of Sigma-Pi neurons[13,45] can implement the recursive binding ("Networks of Sigma-Pi neurons for tensor product and binding" section). While Sigma-Pi neurons are an idealized model there is experimental evidence for multiplication-like nonlinearity by individual nerve cells[46].

For predicting dynamical systems ("Experiments on CPU" section), the performance of our approach is either better than or equal to that of the product representation, echo state network, and multilayer perceptron baselines. Further, it improves the product representation using fewer dimensions with matching performance (e.g., Fig. 4b). Alternatively, higher performance can be attained with moderately increased dimensions to accommodate the features of increased order (Fig. 4c). These results emphasize the role of dimensionality as a tuneable hyperparameter of the proposed approach. Note that this is the only additional hyperparameter introduced beyond the hyperparameters in the product representation scheme (i.e., the choice of delayed states, order of polynomial features, and the regularization parameter)[12]. As follows from the results in Fig. 4, the dimensionality of randomized representations does not require extensive tuning. A simple heuristic is to initially use the number of features in the product representation, a conservative estimate that can often be reduced in practice. Thus, the proposed approach introduces minimal overhead to the hyperparameter search space compared to that of the product representation scheme. In practice, the expected performance increases with increased dimensionality of distributed representations. Dimensionality is a way of controlling the trade-off between the performance and resource-efficiency of the model.

To generate an efficient neuromorphic realization of binding, we utilize how distributed representations can be computed in terms of networks of Sigma-Pi neurons. We further explore randomly

connected Sigma-Pi networks, Supplementary Material S-III, and discuss the trade-offs for different VSA models in "Networks of Sigma-Pi neurons for tensor product and binding" section. Importantly, such compositional distributed representations can be computed by networks of recurrently connected neurons, which further benefit from neuromorphic hardware acceleration. We implement the sparse block code[32,36] on the Loihi 2 neuromorphic chip[15,39] ("Experiments on neuromorphic hardware" section), which is advantageous because of the limited number of synaptic connections required. Notably, the representations computed by the neuromorphic realization performed very close to their CPU counterparts (Fig. 5). It is anticipated that these findings will further impact advances in developing neuro-inspired algorithms, circuits, and applications within the neuromorphic computing community.

## Methods
### Product representation of higher-order features
The product representation of the feature space in Eq. (1) was investigated recently in ref. 12. It is assumed that at each time point $i$ there are $d$ states of the system of interest – $\mathbf{X}(i) \in \mathbb{R}^d$. For the current $i$-th time point, the construction of the feature space begins by forming a vector representing a trajectory of the past observations, which reflects first-order features ("linear" part of the feature space) denoted as $\mathbf{G}^{(1)}(i)$. The current trajectory is specified as a tuple $\mathcal{M}$ that contains indices of $k$ single time points to be included in the trajectory where the indices are spaced $s$ time points apart:

$$\mathcal{M} = (i, i - s, i - 2s, \ldots, i - (k-1)s). \tag{4}$$

Given $\mathcal{M}$, the trajectory is then formed by concatenating the corresponding state vectors:

$$\mathbf{G}^{(1)}(i) = \bigoplus_{l=1}^{k} \mathbf{X}(\mathcal{M}_l) \in \mathbb{R}^{dk}. \tag{5}$$

The resulting feature space could be extended further by considering higher-order features ("nonlinear" part of the feature space) formed from $\mathbf{G}^{(1)}(i)$ using the tensor product in Eq. (7). For example, second-order features $\mathbf{G}^{(2)}(i)$ are constructed as:

$$\mathbf{G}^{(2)}(i) = \mathbf{G}^{(1)}(i) \otimes \mathbf{G}^{(1)}(i). \tag{6}$$

The resulting representation contains $\binom{dk+1}{2} = \frac{dk(dk+1)}{2}$ unique entries in its upper triangular matrix. The explicit feature space can be formed using only these unique entries instead of total $d^2k^2$ entries. In fact, this step has been used in ref. 12 (denoted as $\lceil \otimes \rceil$), here this optimization step is also used for the experiments in "Experiments on CPU" section and elsewhere but we skip the extra notation for the sake of clarity.

The same principle can be used to obtain features of any other higher-order degree. For each problem, one needs to specify the set of desired orders $\mathcal{T} \subset \{0, \ldots, t, \ldots, n-1\}$, where each integer $t+1 \in \mathcal{T}$ describes a desired order of polynomial features to be considered within the feature space. Once $\mathcal{T}$ is specified, by concatenating features of $|\mathcal{T}|$ desired orders, the complete representation is obtained in a single feature vector $\mathbf{G}(i)$:

$$\mathbf{G}(i) = \bigoplus_{t \in \mathcal{T}} \left( \bigotimes^{t} \mathbf{G}^{(1)}(i) \right) = \bigoplus_{t \in \mathcal{T}} \left( \bigotimes^{t} \left( \bigoplus_{l=1}^{k} \mathbf{X}(\mathcal{M}_l) \right) \right), \tag{7}$$

where $\bigotimes^{t}$ denotes the number of times the tensor product is applied to $\mathbf{G}^{(1)}(i)$: $t = 0$ implies first-order features, $t = 1$ results in second-order features, etc. $\mathbf{G}(i)$ may also include an additional optional feature with the constant bias to account for the intercept term. The dimensionality

of the product representation grows exponentially with the highest order of the features and is computed as:

$$\sum_{t \in \mathcal{T}} \binom{dk + t}{t + 1}, \tag{8}$$

As in other machine learning algorithms (e.g., within reservoir computing or kernel machines), the product representation in Eq. (7) is used to produce the prediction for the current time point through the linear transformation specified by a readout matrix (denoted as $\mathbf{W}_{\text{out}}$) following: $\hat{\mathbf{Y}}(i) = \mathbf{W}_{\text{out}}\mathbf{G}(i)$ where the prediction $\hat{\mathbf{Y}}(i)$ is a scalar or vector approximating some desired output such as predicting system's state vector for the next time point $-\hat{\mathbf{Y}}(i) = \hat{\mathbf{X}}(i+1)$. To obtain the readout matrix, the training data in the form of representations for $r$ training time points in matrix $\mathbf{G}$ and the corresponding output values in $\mathbf{Y}$ are used. Given these training data, $\mathbf{W}_{\text{out}}$ can be easily estimated using the ridge regression:

$$\mathbf{W}_{\text{out}} = \mathbf{Y}\mathbf{G}^\top \left(\mathbf{G}\mathbf{G}^\top + \alpha\mathbf{I}\right)^{-1}, \tag{9}$$

where $\mathbf{I}$ is the identity matrix while $\alpha$ is the regularization hyperparameter that controls overfitting to the training data.

## Implicit realization via polynomial kernel machines

We have seen in "Joint representations formed by concatenation or tensor product" section that the similarity (inner product) structure of the space formed by concatenation is of an additive nature while that of the tensor product is of a multiplicative nature. It is, therefore, instructive to mathematically express the effect of concatenation and tensor product on the similarity structure in the resulting feature space. For example, when forming second-order features as in Eq. (6), it can be shown that the inner product between representations defined by two buffers $\mathcal{M}^{(i)}$ and $\mathcal{M}^{(j)}$ is equivalent to:

$$\left\langle \left(\bigoplus_{l=1}^{k} \mathbf{X}(\mathcal{M}_l^{(i)})\right) \otimes \left(\bigoplus_{l=1}^{k} \mathbf{X}(\mathcal{M}_l^{(i)})\right), \left(\bigoplus_{l=1}^{k} \mathbf{X}(\mathcal{M}_l^{(j)})\right) \otimes \left(\bigoplus_{l=1}^{k} \mathbf{X}(\mathcal{M}_l^{(j)})\right) \right\rangle$$
$$= \left\langle \bigoplus_{l=1}^{k} \mathbf{X}(\mathcal{M}_l^{(i)}), \bigoplus_{l=1}^{k} \mathbf{X}(\mathcal{M}_l^{(j)}) \right\rangle^2. \tag{10}$$

First, Eq. (10) makes it evident that the use of the tensor product results in polynomial features[22]. Second, the similarity structure formed by the tensor product is of a self-conjunctive nature and can be expressed as the product of inner products in space formed by concatenation. In general, for $n$th-order features, the inner product is expressed as:

$$\left\langle \bigotimes_{n-1}\left(\bigoplus_{l=1}^{k} \mathbf{X}\left(\mathcal{M}_l^{(i)}\right)\right), \bigotimes_{n-1}\left(\bigoplus_{l=1}^{k} \mathbf{X}\left(\mathcal{M}_l^{(j)}\right)\right) \right\rangle$$
$$= \left\langle \bigoplus_{l=1}^{k} \mathbf{X}\left(\mathcal{M}_l^{(i)}\right), \bigoplus_{l=1}^{k} \mathbf{X}\left(\mathcal{M}_l^{(j)}\right) \right\rangle^n = \left(\sum_{l=1}^{k} \left\langle \mathbf{X}\left(\mathcal{M}_l^{(i)}\right), \mathbf{X}\left(\mathcal{M}_l^{(j)}\right) \right\rangle\right)^n, \tag{11}$$

where Eq. (11) also takes into account the fact that the inner product of concatenation is the sum of the inner products in the spaces being concatenated. Eq. (11) suggests that the same functionality can be expressed through the lens of kernel methods because the feature space in Eq. (7) is identical to the explicit feature map corresponding to the polynomial kernels of various degrees[22]. The kernel functions can be evaluated by simply exponentiating the inner products of the first-order features as in Eq. (11), where the kernel function $\kappa(\mathbf{G}^{(1)}(i), \mathbf{G}^{(1)}(j))$ for Eq. (7) is

computed as:

$$\kappa(\mathbf{G}^{(1)}(i), \mathbf{G}^{(1)}(j)) = \sum_{t \in \mathcal{T}} \left\langle \bigoplus_{l=1}^{k} \mathbf{X}(\mathcal{M}_l^{(i)}), \bigoplus_{l=1}^{k} \mathbf{X}(\mathcal{M}_l^{(j)}) \right\rangle^{t+1}$$
$$= \sum_{t \in \mathcal{T}} \left(\sum_{l=1}^{k} \left\langle \mathbf{X}(\mathcal{M}_l^{(i)}), \mathbf{X}(\mathcal{M}_l^{(j)}) \right\rangle\right)^{t+1}. \tag{12}$$

Given a particular kernel function, entries of the kernel matrix $\mathbf{K} \in \mathbb{R}^{r \times r}$ are obtained from the inner products between the states included in the trajectories of $r$ training time points as:

$$\mathbf{K}_{ij} = \kappa(\mathbf{G}^{(1)}(i), \mathbf{G}^{(1)}(j)). \tag{13}$$

In turn, the kernel matrix $\mathbf{K}$ can be used for obtaining the kernel regression machine that is another form of the readout matrix where the prediction for the $m$-th output state ($\hat{\mathbf{Y}}_m(i)$) is obtained as the weighted sum of the kernel function values between $r$ training time points and the point $i$ to be evaluated. Similar to the readout matrix in Eq. (9), the vector of weights of the training time points (denoted as $\boldsymbol{\alpha}^{(m)}$) could be computed with the ridge regression as:

$$\boldsymbol{\alpha}^{(m)} = (\mathbf{K} + \alpha\mathbf{I})^{-1}\mathbf{Y}_{m:}^\top. \tag{14}$$

The prediction can then be computed as:

$$\hat{\mathbf{Y}}_m(i) = \sum_{j=1}^{r} \boldsymbol{\alpha}_j^{(m)} \kappa(\mathbf{G}^{(1)}(j), \mathbf{G}^{(1)}(i)). \tag{15}$$

An empirical experiment on predicting the Lorenz63 system ("Tasks and experimental configurations" section) comparing the product representation and the corresponding kernel machine is reported in Supplementary Material S-IV. As can be seen from Eq. (15), the kernel machine's size depends on the number of training time points $r$ rather than on the dimensionality of the explicitly constructed product representation. On the one hand, this could be beneficial in the situations when kernel's feature map has large dimensionality while, on the other hand, it could be an issue for large-scale datasets. The latter issue motivated the seminal result in ref. 25 suggesting to use randomized representations for approximating certain kernel functions.

## Randomized representations of higher-order features with binding

We have already considered two ways of realizing the feature space formed by concatenation and tensor product operations: product representation in "Product representation of higher-order features" section and implicit realization with the kernel machine in "Implicit realization via polynomial kernel machine" section. Here, we use the principles of hyperdimensional computing/vector symbolic architectures (VSA)[16–18,30] to introduce the third way of embedding polynomial kernel functions into *randomized distributed representations* that approximate the similarity in the corresponding feature space $\mathbf{G}$, Eq. (7). To manipulate randomized distributed representations of data, VSA defines operations on them such as binding (denoted as $\circ$), permutation (denoted as $\rho$), and superposition (denoted as $+$) that are essential for approximating polynomial kernels. The exact realization of these operations depends on a particular model being used[30,31]. For example, for real-valued representations, there is a holographic reduced representation model[37] that defines the superposition as a component-wise addition while the binding operation is realized via a circular convolution so both operations preserve the dimensionality of representations, $D$ (see more examples in "Networks of Sigma-Pi neurons for tensor product and binding" and "Experiments on neuromorphic hardware" sections). In the context

of this study, the intuition behind the operations is that the permutation randomizes distributed representations, the superposition approximates concatenation in the product representation, and the binding approximates the tensor product as it has been shown in ref. 32.

The initial step in VSA is a transformation of data from their original representation into randomized vector representations (denoted as $\psi(\mathbf{X}(i))$). For concreteness, let us consider a hypothetical system with two observable states $x$ and $y$ ($\mathbf{X}(i) = x(i) \oplus y(i)$, $d = 2$). There are several ways of making a transformation for numeric vectors. A well-known way that is used commonly for randomized neural networks[5] is based on a random projection[47–49] where a $d$-dimensional state vector $\mathbf{X}(i)$ is transformed into a $D$-dimensional vector:

$$\psi(\mathbf{X}(i)) = \mathbf{W}_{\text{in}}\mathbf{X}(i) = x(i)\mathbf{x} + y(i)\mathbf{y}, \tag{16}$$

where $\mathbf{W}_{\text{in}} = [\mathbf{x}; \mathbf{y}] \in \mathbb{R}^{D \times d}$ is a random projection matrix consisting of $d = 2$ columns containing $D$-dimensional i.i.d. random vectors (e.g., $\mathbf{x}$ for $x(i)$). These vectors can be thought to play the role of "keys" in key-value pairs while "value" is the measurement for the state vector to be represented. The representation of state's value is done by scaling the magnitudes of its random vector by the corresponding value, for example: $\psi(x(i)) = x(i)\mathbf{x}$. Due to the usage of random vectors for each state in the trajectory, the transformation can be thought of as making an association between state's value and the position of the corresponding state in the trajectory. Since the superposition operation is used commonly to represent sets or tuples, it allows constructing both the randomized representation of a single time point of the system (e.g., $\psi(\mathbf{X}(i)) = x(i)\mathbf{x} + y(i)\mathbf{y}$) as well as forming the compositional distributed representation of the whole trajectory $\bigoplus_{i=1}^{k} \mathbf{X}(\mathcal{M}_i)$ that is defined by $\mathcal{M}$ (denoted as $\mathbf{F}^{(1)}(i)$). For example, for $k = 2$, and $s = 1$ ($\mathcal{M} = (i, i-1)$):

$$\mathbf{F}^{(1)}(i) = \psi(\mathbf{X}(\mathcal{M}_i) \oplus \mathbf{X}(\mathcal{M}_{i-1})) = \mathbf{W}\left(\bigoplus_{l=1}^{k} \mathbf{X}(\mathcal{M}_l)\right) = \sum_{l=1}^{k} \mathbf{W}_{\rho}^{l-1}(\mathbf{W}_{\text{in}}\mathbf{X}(\mathcal{M}_l))$$

$$= \sum_{l=1}^{k} \rho^{l-1}(\mathbf{W}_{\text{in}}\mathbf{X}(\mathcal{M}_l)) = x(i)\mathbf{x} + y(i)\mathbf{y} + x(i-1)\rho(\mathbf{x}) + y(i-1)\rho(\mathbf{y}), \tag{17}$$

where for the considered case: $\mathbf{W} = [\mathbf{W}_{\text{in}}; \mathbf{W}_{\rho}\mathbf{W}_{\text{in}}] = [\mathbf{W}_{\text{in}}; \rho(\mathbf{W}_{\text{in}})] = [\mathbf{x}; \mathbf{y}; \rho(\mathbf{x}); \rho(\mathbf{y})]$, the role of a mixing matrix $\mathbf{W}_{\rho}$ that is realized via the permutation $\rho(\cdot)$ is to protect the representation of different time steps in the joint representation of the trajectory[32]. Thus, Eq. (17) is the randomized distributed representation corresponding to the concatenation of several time points in the product representation, cf. Eq. (5). A theoretical argument supporting this claim is presented in the next section. The corresponding empirical evaluation is reported in Supplementary Material S-VII.

The distributed representations of higher-order features are obtained from the distributed representation of the trajectory, Eq. (17). Due to the "computing in superposition" property of VSA[18], this step is trivial and uses only the binding and permutation operations that are different from $\rho(\cdot)$. For example, the distributed representation of second-order features that is analogous to applying the tensor product in the explicit feature space ("Joint representations formed by concatenation or tensor product" section) is obtained by binding $\mathbf{F}^{(1)}(i)$ to its own permuted representation:

$$\mathbf{F}^{(2)}(i) = \mathbf{F}^{(1)}(i) \circ \pi\left(\mathbf{F}^{(1)}(i)\right). \tag{18}$$

This is possible since the binding and permutation operations distribute over the superposition so Eq. (18) can be expanded as follows:

$$\begin{aligned}
\mathbf{F}^{(2)}(i) &= \mathbf{F}^{(1)}(i) \circ \pi\left(\mathbf{F}^{(1)}(i)\right) \\
&= (x(i)\mathbf{x} + y(i)\mathbf{y} + x(i-1)\rho(\mathbf{x}) + y(i-1)\rho(\mathbf{y})) \circ \pi(x(i)\mathbf{x} + y(i)\mathbf{y} + x(i-1)\rho(\mathbf{x}) + y(i-1)\rho(\mathbf{y})) \\
&= x(i)^2 \mathbf{x} \circ \pi(\mathbf{x}) + x(i)y(i)\mathbf{x} \circ \pi(\mathbf{y}) + x(i-1)x(i)\pi(\rho(\mathbf{x})) \circ \mathbf{x} + y(i-1)x(i)\pi(\rho(\mathbf{y})) \circ \mathbf{x} \\
&\quad + x(i)y(i)\pi(\mathbf{x}) \circ \mathbf{y} + y(i)^2 \mathbf{y} \circ \pi(\mathbf{y}) + x(i-1)y(i)\pi(\rho(\mathbf{x})) \circ \mathbf{y} + y(i-1)y(i)\pi(\rho(\mathbf{y})) \circ \mathbf{y} \\
&\quad + x(i-1)x(i)\rho(\mathbf{x}) \circ \pi(\mathbf{x}) + x(i-1)y(i)\rho(\mathbf{x}) \circ \pi(\mathbf{y}) + x(i-1)^2 \rho(\mathbf{x}) \circ \pi(\rho(\mathbf{x})) \\
&\quad + x(i-1)y(i-1)\rho(\mathbf{x}) \circ \pi(\rho(\mathbf{y})) + y(i-1)x(i)\rho(\mathbf{y}) \circ \pi(\mathbf{x}) + y(i-1)y(i)\rho(\mathbf{y}) \circ \pi(\mathbf{y}) \\
&\quad + x(i-1)y(i-1)\pi(\rho(\mathbf{x})) \circ \rho(\mathbf{y}) + y(i-1)^2 \rho(\mathbf{y}) \circ \pi(\rho(\mathbf{y}))
\end{aligned} \tag{19}$$

As follows from the expansion in Eq. (19), the result of binding $\mathbf{F}^{(1)}(i)$ to the permuted version of itself is equivalent to the superposition of $dk^2 = 16$ unique randomized representations of second-order monomials of states present in the trajectory $\mathbf{G}^{(1)}(i) = \mathbf{X}(\mathcal{M}_1) \oplus \mathbf{X}(\mathcal{M}_2) = \mathbf{X}(i) \oplus \mathbf{X}(i-1)$. The terms corresponding to the binding of representations of keys (e.g., $\mathbf{x} \circ \pi(\mathbf{x})$, $\mathbf{x} \circ \pi(\mathbf{y})$, etc.) play the role of randomizing the corresponding representations, which allows them co-existing in a compositional randomized distributed representation $\mathbf{F}^{(2)}(i)$, therefore, the inner product between such representations corresponds to the feature map of the second-order polynomial kernel $[x_i^2, \sqrt{2}x_i y_i, \sqrt{2}x_{i-1}x_i, \sqrt{2}y_{i-1}x_i, y_i^2, \sqrt{2}x_{i-1}y_i, \sqrt{2}y_{i-1}y_i, x_{i-1}^2, \sqrt{2}x_{i-1}y_{i-1}, y_{i-1}^2]$. Distributed representations of desired higher-order features $\mathbf{F}^{(t)}(i)$ are obtained in the same manner by binding permuted versions of $\mathbf{F}^{(1)}(i)$ $t-1$ times. This operator is denoted as $\bigcirc^t : \mathcal{H} \times \mathcal{H} \to \mathcal{H}$ and is defined recursively as $\bigcirc^t(\mathbf{F}^{(1)}(i)) = \pi(\bigcirc^{t-1}(\mathbf{F}^{(1)}(i))) \circ \mathbf{F}^{(1)}(i)$ and $\bigcirc^0\left(\mathbf{F}^{(1)}(i)\right) = \mathbf{F}^{(1)}(i)$.

Finally, if the feature space is constructed using features of various orders (specified by $\mathcal{T}$), concatenation would be needed again in the product representation, cf. Eq. (7) but with VSA the joint distributed representation is constructed via the superposition on the corresponding randomized representations:

$$\mathbf{F}(i) = \sum_{t \in \mathcal{T}} \left( \bigcirc^t \left( \sum_{l=1}^{k} \mathbf{W}_{\rho}^{l-1}(\mathbf{W}_{\text{in}}\mathbf{X}(\mathcal{M}_l)) \right) \right) \in \mathbb{R}^D. \tag{20}$$

Since all operations in Eqs. (17)–(20) are dimensionality-preserving, the distributed representation of the feature space is also $D$-dimensional ($D-1$ if the constant bias is included) and can be used in the same way as the product representation in Eq. (7) where the readout matrix for a predictive model is estimated in the same manner as in Eq. (9) but using the distributed representations $\mathbf{F}(i)$ instead of the product ones $\mathbf{G}(i)$:

$$\mathbf{W}_{\text{out}} = \mathbf{Y}\mathbf{F}^{\top}\left(\mathbf{F}\mathbf{F}^{\top} + \alpha\mathbf{I}\right)^{-1}. \tag{21}$$

## Kernel approximation guarantees for randomized representations

As we have alluded to above, the embedding method we have introduced in "Randomized distributed representations of trajectories" section (see also "Randomized representations of higher-order features with binding" section) can be interpreted as generating an approximate feature map for the polynomial kernel. In fact, when the binding operator is the component-wise product, this method coincides with a well-known procedure for approximating polynomial kernels due to ref. 19. In this section, we derive these kernel approximation properties more formally for other realizations of the binding operation.

Let $\kappa_p : \mathbb{R}^d \times \mathbb{R}^d \to \mathbb{R}$ be the homogeneous polynomial kernel of degree $p$ defined on $\mathcal{X} \subset \mathbb{R}^d$ as:

$$\kappa_p(\mathbf{x}, \mathbf{y}) = \langle \mathbf{x}, \mathbf{y} \rangle^p,$$

where $\langle \mathbf{x}, \mathbf{y} \rangle = \mathbf{x}^\top \mathbf{y} = \sum_{j=1}^{d} x_j y_j$ is the standard Euclidean inner product on $\mathbb{R}^d$. Our goal is to design an embedding $\phi_p : \mathcal{X} \to \mathcal{H} \subset \mathbb{R}^D$, such that $\langle \phi_p(\mathbf{x}), \phi_p(\mathbf{y}) \rangle \approx \kappa_p(\mathbf{x}, \mathbf{y})$, in what follows we show that a scheme based on randomized distributed representations can achieve this in expectation, whereupon one may appeal to concentration to argue that the fidelity of approximation is satisfactory for a particular choice of $D$[19].

The generic procedure is as follows. Given an input $\mathbf{x} \in \mathbb{R}^d$, we compute embeddings $\psi_1, \ldots, \psi_p$, where $\psi_t : \mathcal{X} \to \mathcal{H}$, via random linear maps:

$$\psi_t(\mathbf{x}) = \frac{1}{\sqrt{D}} \mathbf{W}^{(t)} \mathbf{x},$$

where $\mathbf{W}^{(t)} \in \mathbb{R}^{D \times d}$ are random projection matrices whose components are drawn i.i.d. from some distribution with zero mean and unit variance. Consequentially, it is the case that:

$$\mathbb{E}\left[ \mathbf{w}_j^{(t)} \mathbf{w}_k^{(t)\top} \right] = \begin{cases} \frac{1}{D} \mathbf{I}_d & \text{if } j = k \\ \mathbf{0}_d & \text{otherwise}, \end{cases}$$

where $\mathbf{w}_j^{(t)}$ denotes the jth row of $\mathbf{W}^{(t)}$ (which we treat as a $d$-dimensional vector for simplicity), $\mathbf{I}_d$ is the $d \times d$ identity matrix, and $\mathbf{0}_d$ is the $d \times d$ dimensional matrix of zeros. We then construct $\phi_p$ by binding together the embeddings $\psi_t(\mathbf{x})$:

$$\phi_p(\mathbf{x}) = \psi_1(\mathbf{x}) \circ \psi_2(\mathbf{x}) \circ \cdots \circ \psi_p(\mathbf{x}).$$

Our basic claim is that, for several different instantiations of $\circ : \mathcal{H} \times \mathcal{H} \to \mathcal{H}$, it is the case that, for any $\mathbf{x}, \mathbf{y} \in \mathcal{X}$:

$$\mathbb{E}[\langle \phi_p(\mathbf{x}), \phi_p(\mathbf{y}) \rangle] = \langle \mathbf{x}, \mathbf{y} \rangle^p = \kappa_p(\mathbf{x}, \mathbf{y}),$$

where the expectation is taken with respect to randomness in $\mathbf{W}^{(1)}, \ldots, \mathbf{W}^{(p)}$. The claim is true for $p = 1$ since, for any $j \in \{1, \ldots, D\}$:

$$\mathbb{E}[\phi_1(\mathbf{x})_j \phi_1(\mathbf{y})_j] = \mathbb{E}[\psi_1(\mathbf{x})_j \psi_1(\mathbf{y})_j] = \mathbb{E}\left[ (\mathbf{w}_j^{(1)\top} \mathbf{x})(\mathbf{w}_j^{(1)\top} \mathbf{y}) \right]$$
$$= \mathbf{x}^\top \mathbb{E}\left[ \mathbf{w}_j^{(1)} \mathbf{w}_j^{(1)\top} \right] \mathbf{y} = \frac{\langle \mathbf{x}, \mathbf{y} \rangle}{D} = \frac{\kappa_1(\mathbf{x}, \mathbf{y})}{D},$$

whereupon:

$$\mathbb{E}[\langle \phi(\mathbf{x}), \phi(\mathbf{y}) \rangle] = \sum_{j=1}^{D} \mathbb{E}[\phi(\mathbf{x})_j \phi(\mathbf{y})_j] = \frac{D \kappa_1(\mathbf{x}, \mathbf{y})}{D} = \kappa_1(\mathbf{x}, \mathbf{y}).$$

We now show this property is satisfied for an arbitrary integer $p \geq 1$ using the following binding operators:

**Component-wise product.** As noted above, the case of component-wise product coincides with the earlier work of [19], but we re-derive their result here for completeness. Let us define, $\circ : \mathcal{H} \times \mathcal{H} \to \mathcal{H}$ to be $\theta \odot \theta'$, where $\odot$ denotes the component-wise product a.k.a. Hadamard product. Now fix some $j \in \{1, \ldots, D\}$, and observe that:

$$\phi_p(\mathbf{x})_j = \prod_{t=1}^{p} \left( \mathbf{w}_j^{(t)\top} \mathbf{x} \right),$$

whereupon:

$$\mathbb{E}[\phi_p(\mathbf{x})_j \phi_p(\mathbf{y})_j] = \mathbb{E}\left[ \prod_{t=1}^{p} (\mathbf{w}_j^{(t)\top} \mathbf{x})(\mathbf{w}_j^{(t)\top} \mathbf{y}) \right]$$
$$= \prod_{t=1}^{p} \mathbf{x}^\top \mathbb{E}\left[ \mathbf{w}_j^{(t)} \mathbf{w}_j^{(t)\top} \right] \mathbf{y}$$
$$= \prod_{t=1}^{p} \mathbf{x}^\top \mathbf{y} = \frac{\langle \mathbf{x}, \mathbf{y} \rangle^p}{D} = \frac{\kappa_p(\mathbf{x}, \mathbf{y})}{D},$$

where we have used the fact that $\mathbf{w}_j^{(1)}, \ldots, \mathbf{w}_j^{(p)}$ are independent to decompose the expectation over the product. Therefore:

$$\mathbb{E}[\langle \phi_p(\mathbf{x}), \phi_p(\mathbf{y}) \rangle] = \kappa_p(\mathbf{x}, \mathbf{y}),$$

as claimed. We make the following remark concerning a potentially more efficient implementation of the scheme above using the permutation operation.

**Remark 1.** The derivation above only requires $p$-wise independence among any set of $\mathbf{w}_j$'s. This could also be achieved by generating $\psi_1(\mathbf{x}) = \mathbf{W}\mathbf{x}$, and then generating subsequent $\psi_i(\mathbf{x})$s via permutation. That is to say, let $\rho(\cdot)$ be a permutation on $[D]$ with cycle time at least $p$. We then set $\psi_2(\mathbf{x}) = \rho(\psi_1(\mathbf{x}))$, $\psi_3(\mathbf{x}) = \rho(\psi_2(\mathbf{x})) = \rho^2(\psi_1(\mathbf{x}))$, and so on.... This strategy allows us to store and compute $\mathbf{W}\mathbf{x}$ only once, which is advantageous computationally and in terms of memory.

**Remark 2.** The calculations above describe what happens in expectation over randomness in the draw of the $\mathbf{w}_j$. However, it is also possible to provide high-probability bounds on the approximation error from a specific instantiation. This question is analyzed at length in[19], who show that (via their Lemma 8 and Hoeffding's inequality) for any fixed but arbitrary pair of points $\mathbf{x}, \mathbf{y}$, and any $\epsilon > 0$, to guarantee that:

$$|\phi_p(\mathbf{x}) \cdot \phi_p(\mathbf{y}) - \kappa_p(\mathbf{x}, \mathbf{y})| \leq \epsilon,$$

holds with high-probability over randomness in the draw of $\mathbf{w}_1, \ldots, \mathbf{w}_p$, it suffices to choose:

$$D = O((pR/\epsilon)^2),$$

where $R = \max\{\|\mathbf{x}\|_1, \|\mathbf{y}\|_1\}$. The crucial insight is that $D$ depends just quadratically on the maximum polynomial degree $p$, as opposed to exponentially in the explicit case (albeit at the expense of only achieving an approximation to the true kernel).

**Circular convolution.** Let us take $\circ : \mathcal{H} \times \mathcal{H} \to \mathcal{H}$ to be the discrete, circular convolution operator $\circledast$, defined as[37]:

$$(\theta \circledast \theta')_j = \sum_{i=0}^{D-1} \theta_i \theta'_{j-i},$$

where we think of the first component in $\theta$ as having an index of 0, and all subscripts are modulo $D$. As noted above, the claim is trivially true for $p = 1$, and we proceed by induction on $p$. Let us suppose that, for any $p > 1$:

$$\mathbb{E}[\langle \phi_{p-1}(\mathbf{x}), \phi_{p-1}(\mathbf{y}) \rangle] = \kappa_{p-1}(\mathbf{x}, \mathbf{y}).$$

Let us fix some $j \in \{1, \ldots, D\}$. By definition of $\circledast$, we have that:

$$\phi_p(\mathbf{x})_j = \left( \phi_{p-1}(\mathbf{x}) \circledast \psi_p(\mathbf{x}) \right)_j = \sum_{i=0}^{D-1} \phi_{p-1}(\mathbf{x})_i \psi_p(\mathbf{x})_{j-i}.$$

Thus:

$$\phi_p(\mathbf{x})_j \phi_p(\mathbf{y})_j = \left( \sum_{i=0}^{D-1} \phi_{p-1}(\mathbf{x})_i \psi_p(\mathbf{x})_{j-i} \right) \left( \sum_{k=0}^{D-1} \phi_{p-1}(\mathbf{y})_k \psi_p(\mathbf{y})_{j-k} \right)$$

$$= \sum_{i=0}^{D-1} \sum_{k=0}^{D-1} \phi_{p-1}(\mathbf{x})_i \phi_{p-1}(\mathbf{y})_i \left( \mathbf{w}_{j-i}^{(p)\top} \mathbf{x} \right) \left( \mathbf{w}_{j-k}^{(p)\top} \mathbf{y} \right)$$

$$= \sum_{i=k=0}^{D-1} \phi_{p-1}(\mathbf{x})_i \phi_{p-1}(\mathbf{y})_k \left( \mathbf{x}^\top \mathbf{w}_{j-i}^{(p)} \mathbf{w}_{j-k}^{(p)\top} \mathbf{y} \right)$$

$$+ \sum_{i \neq k=0}^{D-1} \phi_{p-1}(\mathbf{x})_i \phi_{p-1}(\mathbf{y})_k \left( \mathbf{x}^\top \mathbf{w}_{j-i}^{(p)} \mathbf{w}_{j-k}^{(p)\top} \mathbf{y} \right).$$

Let us focus first on the second summation. By independence in the draws of $\mathbf{W}^{(1)}, \ldots, \mathbf{W}^{(p)}$:

$$\mathbb{E} \left[ \sum_{i \neq k=0}^{D-1} \phi_{p-1}(\mathbf{x})_i \phi_{p-1}(\mathbf{y})_k \left( \mathbf{x}^\top \mathbf{w}_{j-i}^{(p)} \mathbf{w}_{j-k}^{(p)\top} \mathbf{y} \right) \right] = \cdots$$

$$= \sum_{i \neq k=0}^{D-1} \mathbb{E} \left[ \phi_{p-1}(\mathbf{x})_i \phi_{p-1}(\mathbf{y})_k \right] \mathbb{E} \left[ \mathbf{x}^\top \mathbf{w}_{j-i}^{(p)} \mathbf{w}_{j-k}^{(p)\top} \mathbf{y} \right] = 0,$$

since $\mathbb{E}[\mathbf{w}_{j-i}^{(p)} \mathbf{w}_{j-k}^{(p)\top}] = \mathbf{0}_d$ for $i \neq k$, and, thus, the second summation vanishes in expectation. Turning our attention to the first summation, by independence and the inductive assumption, we have:

$$\mathbb{E} \left[ \sum_{i=k=0}^{D-1} \phi_{p-1}(\mathbf{x})_i \phi_{p-1}(\mathbf{y})_k \left( \mathbf{x}^\top \mathbf{w}_{j-i}^{(p)} \mathbf{w}_{j-k}^{(p)\top} \mathbf{y} \right) \right] = \sum_{i=k=0}^{D-1} \mathbb{E} \left[ \phi_{p-1}(\mathbf{x})_i \phi_{p-1}(\mathbf{x})_k \right] \mathbb{E} \left[ \mathbf{x}^\top \mathbf{w}_{j-i}^{(p)} \mathbf{w}_{j-k}^{(p)\top} \mathbf{y} \right]$$

$$= \frac{\mathbb{E}[\langle \phi_{p-1}(\mathbf{x}), \phi_{p-1}(\mathbf{y}) \rangle] \langle \mathbf{x}, \mathbf{y} \rangle}{D}$$

$$= \frac{\langle \mathbf{x}, \mathbf{y} \rangle^{p-1} \langle \mathbf{x}, \mathbf{y} \rangle}{D} = \frac{\langle \mathbf{x}, \mathbf{y} \rangle^p}{D}$$

Whereupon, we conclude that once again:

$$\mathbb{E}[\langle \phi_p(\mathbf{x}), \phi_p(\mathbf{y}) \rangle] = \kappa_p(\mathbf{x}, \mathbf{y}).$$

## Experiments with traditional reservoir computing networks

Since the literature on reservoir computing is vast, we do not introduce it in detail here (refer, e.g., to refs. [4,40]) but for the sake of completeness we provide the equation for the evolution of the reservoir dynamics of echo state networks that are used as a baseline in the experimental evaluation:

$$\mathbf{F}(i) = \tanh(\beta \mathbf{W}_{in}[1 \oplus \mathbf{X}(i)] + \gamma \mathbf{W} \mathbf{F}(i-1)), \qquad (22)$$

where $\mathbf{F}(i)$ is the $D$-dimensional state of the reservoir at time point $i$. $\mathbf{W}_{in} \in \mathbb{R}^{D \times d+1}$ is an input projection matrix whose components are drawn uniformly from $[-1, 1]$, $\mathbf{W} \in \mathbb{R}^{D \times D}$ is a mixing random reservoir connectivity matrix whose components are drawn i.i.d. from the standard normal distribution and then $\mathbf{W}$ is normalized so that its spectral radius is one (to achieve "echo-state property"[3]). The contribution of the present state of the system $\mathbf{X}(i)$ is controlled by the projection gain hyperparameter $\beta$ while the contribution of the previous states of the reservoir is controlled by $\gamma$ – the spectral radius of $\mathbf{W}$.

## Tasks and experimental configurations

**Tasks and configurations of feature spaces.** To evaluate the distributed representation of the feature space and compare it to the product representation, in the first place, we followed the experimental protocol of study[12] that involved three tasks as well as included an additional task that was demonstrated to be challenging for the product representation in ref. [38] and a task involving a dynamical system with many input channels. For all tasks, the readout matrix $\mathbf{W}_{out}$ is obtained using the ridge regression following Eqs. (9) or (14).

The first and fifth tasks are designed using time-series generated by numerically integrating a system developed by Lorenz in 1963[50]. The system includes three coupled nonlinear differential equations (referred to as Lorenz63):

$$\dot{x} = 10(y-x), \quad \dot{y} = x(28-z) - y, \quad \dot{z} = xy - 8z/3, \qquad (23)$$

thus, system's state at time point $i$ is characterized by vector $\mathbf{X}(i) \equiv [x(i), y(i), z(i)]^\top$. The system in Eq. (23) forms a strange chaotic attractor so it displays deterministic chaos that is sensitive to initial conditions (Lyapunov time is 1.1-time units). The system is sampled at $dt = 0.025$ and models are trained on $r = 400$ time points that is about ten Lyapunov times. Supplementary Material S-IX reports the results of experiments with varying amounts of training data in the range $r \in [160, 960]$.

The second system forming a strange chaotic attractor that is used for the second task is a double-scroll electronic circuit[51]. It is described by (system's state is $\mathbf{X}(i) \equiv [V_1(i), V_2(i), I(i)]^\top$):

$$\dot{V}_1 = V_1/R_1 - \Delta V/R_2 - 2I_r \sinh(\beta \Delta V), \quad \dot{V}_2 = \Delta V/R_2 + 2I_r \sinh(\beta \Delta V)$$
$$- I, \quad \dot{I} = V_2 - R_4 I \qquad (24)$$

where $\Delta V = V_1 - V_2$; while the parameters are set to $R_1 = 1.2$, $R_2 = 3.44$, $R_4 = 0.193$, $\beta = 11.6$, and $I_r = 2.25 \times 10^{-5}$ resulting in a Lyapunov time of 7.8-time units. To account for slower Lyapunov time, the system is sampled at $dt = 0.25$ but models are still trained on $r = 400$ time points.

The third system is the Mackey–Glass[52] that is used frequently within, e.g., the reservoir computing literature as a showcase[3,53]. It is formulated using the following time-delay differential equation:

$$\dot{u}(t) = \beta \frac{u(t-\tau)}{1 + u(t-\tau)^n} - \gamma u(t), \qquad (25)$$

where the parameters are set to $\beta = 0.2$, $\gamma = 0.1$ $\tau = 17$, and $n = 10$ (Lyapunov time is about 185-time units). Similar to the above systems, the training data spans about ten Lyapunov times but the system is sampled at $dt = 3.0$ resulting in $r = 600$ training time points. The number of training points is higher than for the two systems above because increasing $dt$ further substantially worsens the predictive performance.

The fourth system is the Kuramoto–Sivashinsky system[54] that was used to introduce the parallel reservoir computing scheme[55]. It describes the slow variation of the vibration function of a system extended in the space with the fourth-order partial differential equation:

$$\dot{u}(t) = -u\dot{u}(x) - \ddot{u}(x^2) - u^{....}(x^4), \qquad (26)$$

where the scalar field $y(x, t)$ is periodic in the interval $[0, \mathcal{L}]$; $\mathcal{L} = 22$ in this study (Lyapunov time is about 20-time units). In contrast to the above systems, the training data had to be increased substantially so it spans one hundred Lyapunov times. The system is integrated on a grid of sixteen equally spaced points ($d = 16$) and is sampled at $dt = 0.5$ resulting in $r = 4000$ training time points.

The first three tasks that were reported in "Experiments on CPU" section of the main text while the fourth task is reported in Supplementary Material S-I using the time-series produced by the above systems in Eqs. (23)–(26) to predict their dynamics using one-step-ahead prediction. In particular, the readout matrices $\mathbf{W}_{out}$ for the models are trained to predict the difference between the current and the next states of the system ($\mathbf{Y}(i) = \mathbf{X}(i+1) - \mathbf{X}(i)$) using either baseline models, product representation ("Product representation of higher-order features" section) or distributed representation ("Randomized

representations of higher-order features with binding" section) of the feature space that are formed from $k$ single time points of the system (i.e., the trajectory) that are parameterized by $k$-tuple $\mathcal{M}$ where indices within the tuple are specified relative to the current time point $i$. For the sake of fair comparison with the product representation, the experiments with the distributed representation are performed with the idealized memory buffer that stores exactly $k$ single time points.

During the training phase, states of the system and the corresponding target values are provided as the ground truth (i.e., teacher forcing[56]). For the first four tasks, during the prediction phase, the model's prediction at the $i$-th time point is used as an input for the $i+1$-th time point so the model operates in the autoregressive mode, i.e., autonomously unfolding in time without any external signal such that $\hat{\mathbf{X}}(i+1) = \hat{\mathbf{X}}(i) + \mathbf{W}_{\text{out}} \mathbf{G}(i)$ or $\hat{\mathbf{X}}(i+1) = \hat{\mathbf{X}}(i) + \mathbf{W}_{\text{out}} \mathbf{F}(i)$.

For predicting the Lorenz63 system, the feature space includes constant bias (set to 1), first- as well second-order features, $\mathcal{T} = (0, 1)$, that for the product representation is obtained as:

$$\mathbf{G}(i) = 1 \oplus \mathbf{G}^{(1)}(i) \oplus \mathbf{G}^{(2)}(i), \qquad (27)$$

where the representation includes $1 + dk + \frac{dk(dk+1)}{2}$ features, which given the concrete values of hyperparameters: $d = 3, k = 2, \mathcal{M} = (i, i-1)$, results in 28 dimensions in $\mathbf{G}(i)$.

The corresponding randomized distributed representation is obtained as:

$$\mathbf{F}(i) = 1 \oplus [\mathbf{F}^{(1)}(i) + \mathbf{F}^{(2)}(i)] = 1 \oplus [\mathbf{F}^{(1)}(i) + \mathbf{F}^{(1)}(i) \circ \rho(\mathbf{F}^{(1)}(i))], \qquad (28)$$

where the representation is $D$-dimensional. Supplementary Material S-V shows the results of experiments with other configurations of $\mathcal{T}$.

For predicting the double-scroll system, third-order features are used, $\mathcal{T} = (0, 2)$, due to attractor's odd symmetry so that the product representation is formed as:

$$\mathbf{G}(i) = \mathbf{G}^{(1)}(i) \oplus \mathbf{G}^{(3)}(i), \qquad (29)$$

which has $dk + \frac{dk(dk+1)(dk+2)}{6}$ dimensions, which given the concrete values of hyperparameters: $d = 3, k = 2, \mathcal{M} = (i, i-1)$ results in 62 dimensions in $\mathbf{G}(i)$.

The corresponding $D$-dimensional randomized distributed representation is formed as:

$$\mathbf{F}(i) = 1 \oplus [\mathbf{F}^{(1)}(i) + \mathbf{F}^{(3)}(i)] = 1 \oplus [\mathbf{F}^{(1)}(i) + \mathbf{F}^{(1)}(i) \circ \rho(\mathbf{F}^{(1)}(i)) \circ \rho^2(\mathbf{F}^{(1)}(i))], \qquad (30)$$

where the constant bias is also not necessary as for the Lorenz63 system but is used for the sake of consistency with the models for other tasks.

For the Mackey–Glass system, features up to third-order are used in the simplest product representation, $\mathcal{T} = (0, 1, 2)$:

$$\mathbf{G}(i) = 1 \oplus \mathbf{G}^{(1)}(i) \oplus \mathbf{G}^{(2)}(i) \oplus \mathbf{G}^{(3)}(i), \qquad (31)$$

with $1 + dk + \frac{dk(dk+1)}{2} + \frac{dk(dk+1)(dk+2)}{6}$ features. In the Mackey–Glass system, there is only one observable state so $d = 1$ but $k$ is set to 6 (see Chapter 5.4 in ref. 38 for details of choosing the delay taps) producing 84 features for $\mathbf{G}(i)$.

For the distributed representation, this task is used to demonstrate that it dissects the strict dependency between $D$ and the number of polynomial features; this approach, therefore, investigates several configurations starting from the representation that includes features

up to fourth-order, $\mathcal{T} = (0, 1, 2, 3)$:

$$\mathbf{F}(i) = 1 \oplus [\mathbf{F}^{(1)}(i) + \mathbf{F}^{(2)}(i) + \mathbf{F}^{(3)}(i) + \mathbf{F}^{(4)}(i)]. \qquad (32)$$

and all the way up to representations with features of up to order seven.

For the Kuramoto–Sivashinsky system, features up to third-order are used in the product representation, $\mathcal{T} = (0, 1, 2)$:

$$\mathbf{G}(i) = 1 \oplus \mathbf{G}^{(1)}(i) \oplus \mathbf{G}^{(2)}(i) \oplus \mathbf{G}^{(3)}(i), \qquad (33)$$

with $1 + dk + \frac{dk(dk+1)}{2} + \frac{dk(dk+1)(dk+2)}{6}$ features. Given sixteen observable states $d = 16$ and $k = 2$, $\mathbf{G}(i)$ includes 6, 545 features.

The corresponding $D$-dimensional randomized distributed representation is formed as:

$$\mathbf{F}(i) = 1 \oplus [\mathbf{F}^{(1)}(i) + \mathbf{F}^{(2)}(i) + \mathbf{F}^{(3)}(i)]. \qquad (34)$$

Finally, the fifth task also uses the Lorenz63 system to imitate the scenario where not all observable states are available upon the deployment of the model so some missing data needs to be predicted. During the training phase, all three states are observed where values of state $z$ at $i$-th time point are used as the ground truth ($\mathbf{Y}(i) = z(i)$) for one-step-ahead prediction from the previously observed values of $x$ and $y$ states. Thus, during the prediction phase, the model is only observing $x$ and $y$ states and tries to infer the value of $z$ at the current time point. As in the first task, here, first- and second-order features, $\mathcal{T} = (0, 1)$, are sufficient to obtain predictions of high quality. For the sake of brevity of the main text, the results for this task are reported in Supplementary Material S-II.

**Choice of hyperparameters.** The first half of this section introduced the considered tasks and the corresponding configurations for constructing the feature spaces for the product and distributed representations. Upon defining the construction of the feature space, there are at most two hyperparameters remaining: (1) the regularization parameter $\alpha$ for obtaining the readout matrix, and (2) the dimensionality, $D$, of the randomized representations. Note that (2) is only necessary for the distributed representation. For both approaches, the choice of the regularization parameter was performed separately for each task. The grid search was in the range $\alpha \in \{1 \times 10^{-12}, 1 \times 10^{-11}, ..., 1\}$ where for each value in the range 1000 randomly initialized realizations of the considered systems were evaluated (100 for the Kuramoto–Sivashinsky system due to the computing costs). For each realization, the normalized root-mean-square error ("Performance metrics" section) was computed over three Lyapunov times. The value resulting in the smallest median error was chosen as the optimal one. The optimal values of $\alpha$ are indicated next to each specific experiment. For the distributed representation, while searching for optimal $\alpha$, the value of $D$ was fixed to the dimensionality of the corresponding product representation, Eq. (8). The same $\alpha$ was used when conducting experiments involving varying $D$.

Since the feature space in echo-state networks is defined implicitly, a larger hyperparameter search is needed for proper comparison. In addition to the dimensionality of the reservoir, $D$, and regularization parameter, $\alpha$, we also considered the projection gain of the input $\beta$ and the spectral radius of the reservoir connectivity matrix $\gamma$, cf. Eq. (22). As for the distributed representation, while performing the grid search over $\alpha$, $\beta$, and $\gamma$, $D$ was set to match the configuration of the product representation for the considered task. For the grid search, the range of $\alpha$ was the same as above while for $\beta$ and $\gamma$, we followed the configuration from ref. 57 with seven points for each parameter that were distributed evenly in the range [0.1, 1]. Thus, for each task, 637 different configurations of hyperparameters were considered for the echo-state network. As with the other approaches, the best

configuration was chosen based on the median normalized root-mean-square error across 1000 different realizations of the considered systems (100 for the Kuramoto–Sivashinsky system).

### Performance metrics

As a way to measure the quality of predictions for the experiments in the main text and Supplementary Material, a standard metric is used – normalized root-mean-square error (NRMSE). Given the ground truth $\mathbf{Y} \in \mathbb{R}^{m \times r}$ with $m$ output states & $r$ evaluation samples as well as the corresponding predictions in $\hat{\mathbf{Y}}$, NRMSE is computed as:

$$\text{NRMSE} = \sqrt{\frac{\sum_{i=1}^{r} \sum_{j=1}^{m} \left( \mathbf{Y}_j(i) - \hat{\mathbf{Y}}_j(i) \right)^2}{mr \sum_{j=1}^{m} \sigma_{\mathbf{Y}_{j:}}^2}}, \tag{35}$$

where $\sigma_{\mathbf{Y}_{j:}}^2$ denotes the empirical variance (i.e., the variance calculated numerically from the data) of the $j$th output state.

## Data availability

The data that support the plots within this study and other findings can be generated using the available code and data available online in the Code Ocean at https://doi.org/10.24433/CO.7208482.v2.

## Code availability

The computer code and/or data used to produce the results reported in the study accompany this article and are deposited in the Code Ocean; available online at https://doi.org/10.24433/CO.7208482.v2.

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

## Acknowledgements

D.K. has received funding from the European Union's Horizon 2020 research and innovation programme under the Marie Skłodowska-Curie grant agreement No 839179. The work of C.J.K. was supported by the Department of Defense (DoD) through the National Defense Science & Engineering Graduate (NDSEG) Fellowship Program. The work of C.J.K. and B.A.O. was supported by the Center for the Co-Design of Cognitive Systems (CoCoSys), one of seven centers in JUMP 2.0, a Semiconductor Research Corporation (SRC) program sponsored by DARPA, as well as NSF Awards 2147640 and 2313149. The work of D.K., B.A.O., and F.T.S. was supported in part by Intel's THWAI program. F.T.S. was supported by NSF Grant IIS1718991, NIH Grant R01-EB026955 and by the Kavli Foundation. The authors acknowledge the EuroHPC Joint Undertaking for awarding this study access to the EuroHPC supercomputer LUMI (project No 465000448), hosted by CSC (Finland) and the LUMI consortium through a EuroHPC Regular Access call.

## Author contributions

All authors participated in discussions shaping the ideas, brainstorming the experiments, and defining the research questions in this study. D.K. proposed the mathematical formulation of distributed representations for higher-order features and implemented the experiments evaluating the proposed approach. D.K., C.J.K., A.T., B.A.O., F.T.S., and E.P.F. participated in the design of Sigma-Pi networks. E.P.F. implemented the Sigma-Pi networks on Loihi 2 and ran the corresponding experiments. A.T. proved that the distributed representation approximates the polynomial kernel and wrote the corresponding parts of the manuscript. E.P.F., F.T.S., and B.A.O. advised the project. The manuscript was written with input from all authors.

## Funding

## Competing interests

The authors declare no competing interests.
