## [Transparent Peer Review file · Nature Communications]

Principled Neuromorphic Reservoir Computing

Corresponding Author: Dr Denis Kleyko

Version 0:

Reviewer comments:

Reviewer #1

(Remarks to the Author)

In the manuscript "Principled Neuromorphic Reservoir Computing", the authors develop a new configurable neuromorphic representation for the next generation reservoir computing framework based on the higher-order polynomial features. Specifically, they utilize the principles of the vector symbolic architectures to compute the randomized distributed representations of the higher-order polynomial kernels, which can reduce the dimensionality of the feature space compared with the original product representations. These features make it easier to deploy on hardware devices.

Though the results look interesting, there are several major and minor questions regarding this article.

1. Based on my understanding, I wonder if the proposed framework simply employed a lower-dimensional feature of the original next generation RC.
2. Previous work [1] has demonstrated that under certain conditions of parameter randomness, conventional Reservoir Computing (RC) converges to a unique recurrent kernel as the number of the intermediate neurons tends to infinity. A finite-dimensional RC can be seen as a random feature approximation of this kernel. In this paper, the authors choose a specific kernel, namely the polynomial kernel, and utilize its random features as the high-dimensional representation of RC. It is essential to further explain why this specific choice might be advantageous compared to the broader recurrent kernels corresponding to conventional infinite-dimensional RC or random-feature-based RC corresponding to some other specific kernels. Additionally, a performance comparison between this method and other approaches [1] based on recurrent kernels and random features in prediction tasks should be provided.
3. The experimental comparisons lack the baseline model using the conventional RC framework. Additionally, since the model introduces time-delay terms as the input, it should also be compared with some delay-based RC frameworks [2].
4. The experiments conducted in this paper are primarily on relatively simple toy models, which do not adequately demonstrate the effectiveness and broad applicability of the method. Can the predictive capabilities of this method be showcased on high-dimensional systems (such as the Kuramoto-Sivashinsky (KS) chaotic system) and systems with observational noise?
5. Actually, the authors employ the delay embedding (see Eq. (4)), and the authors may discuss the difference between the proposed framework and Taken's Embedding theory.
6. Since the authors did not provide the detailed experimental configurations (a major concern), it is difficult to validate the correctness and generality of the results, and thus the value of this work is limited. Additionally, this work lacks ablation studies on the influences of operations such as superposition, binding, and permutation. The robustness (e.g., random seed, noise) and hyperparameters (e.g., order) should be sufficiently evaluated.
7. Though the authors introduce circular convolution, I wonder if this operator is necessary in the framework from both theoretical and experimental aspects.
8. There is no evidence to support the claim "...is to protect representation of different time steps in the joint representation of the trajectory".

9. Some symbols in the paper are reused, causing confusion. For example, the 'i' in Eq.(5, 7, 10, 11, ...) represents the time index of G, while also denotes the elements in the set M. The bold font of the first term in the equation after the sentence '... is defined recursively as' is not correct.

References:

[1] Dong J, Ohana R, Rafayelyan M, et al. Reservoir computing meets recurrent kernels and structured transforms[J]. *Advances in Neural Information Processing Systems*, 2020, 33: 16785-16796.

[2] Duan X Y, Ying X, Leng S Y, et al. Embedding theory of reservoir computing and reducing reservoir network using time delays[J]. *Physical Review Research*, 2023, 5(2): L022041.

(Remarks on code availability)

Reviewer #2

(Remarks to the Author)

In their manuscript "Principled Neuromorphic Reservoir Computing," Kleyko and co-authors propose what might be considered a third-generation form of reservoir computing, combining the randomized distributed representations of traditional reservoir computing (RC) with the higher-order input features used to great effect in by a nonlinear vector autoregression (NVAR) machine (termed "NG-RC" in Ref. 11 and "product representation" in the current work.). In this way, the authors' "distributed representation" reservoir attained comparable or better NMSEs on classic RC benchmarks using up to or fewer input features compared with the product representation method. The dimension of the input feature space was changed to be a tunable parameter by recognizing that distributed representations incorporating vector symbolic architectures (VSA) operations approximate the same similarity structure of product representations, notably through the superposition (or collective state computing) properties of VSA. Further, the authors propose and demonstrate a neural network implementation of this distributed representation reservoir of Sigma-Pi neurons on Intel's Loihi2 hardware, whose simulation code is available on github (though not the LAVA-VSA toolbox). This demonstration highlighted the power of VSA operators for preserving dimensionality unlike tensor products, such that neural subnets can be trivially reused.

Figure 1 does an excellent job illustrating the key differences and similarities among traditional RC, product representation, and distributed representation approaches. Unlike the NG-RC, which could be argued as strictly better than traditional RC for prediction in terms of parameter tuning and training data efficiency, the distributed representation did not uniformly perform better than the NG-RC (product representation), though it was at minimum comparable. A critical contribution of this paper then is also to unify the mathematical descriptions of each approach, thereby, providing a more formal explanation for the comparable behavior via different means. As the authors' note, the product representation's feature size was often a good initial estimate for the distributed representation.

The manuscript is very well written, though dense and detailed, as necessary to clearly detail the mathematical underpinnings. I particularly appreciate the expanded equations in section 4 to ensure the reader could follow the terms intended by the notation. Due to the popularity of RC and the advantages of NG-RC (product representation), this distributed representation and mathematical interpretation will be of great interest to the RC community and readers of *Nature Communications*. Aside from minor questions below, I recommend publication.

1) As far as I understand, the results from Fig. 3 were not computed on the Intel Loihi2 hardware, only Fig. 5 results. Section 2.6 is described as an ablation study, but since the authors' highlight the size of the Intel Hala Point, it is noticeable that they chose to showcase this large-scale hardware on an ostensibly simpler problem (Fig. 5). It would seem the neuromorphic hardware would be large enough to perform the chaotic time series prediction tasks (Fig. 3). Did the authors try to replicate the results on Fig. 3 as well on the Hala Point?

2) The NG-RC paper (Ref. 11) argued that for traditional RC, multiple reservoirs had to be created and parameters tuned to find an optimal RC solution. Their proposed higher-order input thus skipped the variability of the input and reservoir weights entirely, simplifying training. Yet, in this paper, the authors claim that the distributed representation sufficiently approximates the kernel regardless of the random vectors used, thereby, requiring minimal tuning parameters, i.e. the input feature size D and regularization parameter α . It may be worth a remark about the optimization search spaces between the two methods.

Typos:

T1) Section 2.3 "and, in essence, form"

Suggestions:

S1) The Fig. 5 caption or legend should include a remark about the dashed line being CPU results.

S2) The Fig. S.2 caption or legend should include a remark about the grey dash-dot line being structured results. In the spirit of random connections with reservoir computing, I found this insight particularly interesting.

(Remarks on code availability)

I was able to see the code, but I did not run it. I have worked enough with reservoir computing and NG-RC code, that I am confident I can recreate the key paper results from the equations provided in the paper itself.

Version 1:

Reviewer comments:

Reviewer #1

(Remarks to the Author)

In this revised manuscript, the authors have substantially addressed my major concerns in the first round of review. Also it seems that the authors have satisfactorily responded to the comments from the other expert. Now, I therefore suggest the Editor to accept this work.

(Remarks on code availability)

Reviewer #2

(Remarks to the Author)

The authors have satisfactorily answered all my questions, and I myself find their replies to the other reviewer's questions also satisfactory. I also commend the authors' significant expansion of the supplementary materials which I expect will be of substantial interest to this community.

I recommend this manuscript for publication.

(Remarks on code availability)

Revision

Reviewer 1

In the manuscript “Principled Neuromorphic Reservoir Computing”, the authors develop a new configurable neuromorphic representation for the next generation reservoir computing framework based on the higher-order polynomial features. Specifically, they utilize the principles of the vector symbolic architectures to compute the randomized distributed representations of the higher-order polynomial kernels, which can reduce the dimensionality of the feature space compared with the original product representations. These features make it easier to deploy on hardware devices.

We would like to thank the reviewer for accurately characterizing the essence of our study. We appreciate the overall positive feedback and the suggestions for improving the manuscript as well as our experiments. In the following we list comments and manuscript revisions addressing the individual comments.

Though the results look interesting, there are several major and minor questions regarding this article.

- 1. Based on my understanding, I wonder if the proposed framework simply employed a lower-dimensional feature of the original next generation RC.**

Indeed, the proposed randomized representations can approximate polynomial features of the same order as the original next generation reservoir computing. We go beyond the work by [Gauthier et al., 2021] by explaining how to engineer further higher-order polynomial features without the dimensionality growing exponentially within the randomized framework. The feature approximation preserves the polynomial kernel similarity structure between inputs up to a small additive distortion. In addition, we demonstrate how to instantiate polynomial kernels efficiently with Sigma-Pi neurons.

- 2. Previous work [Dong et al., 2020] has demonstrated that under certain conditions of parameter randomness, conventional Reservoir Computing (RC) converges to a unique recurrent kernel as the number of the intermediate neurons tends to infinity. A finite-dimensional RC can be seen as a random feature approximation of this kernel.**

Indeed, our results in Section 4.4 also shows that the proposed approach converges to a unique polynomial kernel as the reservoir’s dimension increases. Here, however, the polynomial kernel is the result of binding operations in the network, which are not present in the reservoir computing framework considered in [Dong et al., 2020]. Our framework does not just claim that there exists a recurrent kernel, but it further illustrates how network motifs can be combined to design a specific kernel.

In this paper, the authors choose a specific kernel, namely the polynomial kernel, and utilize its random features as the high-dimensional representation of

RC. It is essential to further explain why this specific choice might be advantageous compared to the broader recurrent kernels corresponding to conventional infinite-dimensional RC or random-feature-based RC corresponding to some other specific kernels.

It is our perception that (at least) outside of the reservoir computing community, the importance of polynomial features/kernels is already firmly established. Polynomial kernel machines and polynomial regression are widely known and useful tools in machine learning. The earlier results in [Pyle et al., 2021, Gauthier et al., 2021] and our results enrich the repertoire of reservoir computing networks, by explicitly linking [next generation] reservoir networks to the polynomial kernel. The theoretical connection between Volterra series and polynomial kernel regression [Franz and Schölkopf, 2006] supports the idea of using these representations for learning dynamical systems. Practically, the performance advantages of using polynomial higher-order features have already been demonstrated in the prior reservoir computing literature [Pyle et al., 2021, Gauthier et al., 2021, Barbosa and Gauthier, 2022] (we discuss this point further in item (5)). We have revised the second paragraph in the discussion to address these points:

Polynomial kernel machines and polynomial regression are widely known and useful tools in machine learning. The earlier results in [Pyle et al., 2021, Gauthier et al., 2021] and our results enrich the repertoire of reservoir computing networks, by explicitly linking reservoir networks to the polynomial kernels. The theoretical connection between Volterra series and polynomial kernel regression [Franz and Schölkopf, 2006] further supports the idea of using these representations for learning dynamical systems.

Additionally, a performance comparison between this method and other approaches [Dong et al., 2020] based on recurrent kernels and random features in prediction tasks should be provided.

Due to the usage of structured matrices, the structured reservoir computing framework from [Dong et al., 2020] requires less computations for evolving the network than the conventional reservoir computing. But similar to the delay-based reservoir computing framework [Duan et al., 2023] mentioned below in item (3), structured reservoir computing achieves the same performance as the conventional reservoir computing. This means that we would expect the performance curves in Figures 4, S.1, S.2, and S.3 would match those of the conventional reservoir computing.

- 3. The experimental comparisons lack the baseline model using the conventional RC framework. Additionally, since the model introduces time-delay terms as the input, it should also be compared with some delay-based RC frameworks [Duan et al., 2023].**

Following your suggestion, we added a baseline comparison to the conventional reservoir computing framework in both Figure 4 in the main text and in the Supplementary Material (Figures S.1, S.2, and S.3). Thank you for pointing us to the delay-based reservoir computing framework in [Duan et al., 2023]. The authors in [Duan et al., 2023] show that “delayed observables of the RC state, seen as additional non-linear observables, have the same computational power in the system reconstruction” (emphasis added) and conclude that “the numbers of neurons and of time lags can be traded off mutually.” Thus, the curve of the method in [Duan et al., 2023] would coincide with the curve of conventional reservoir computing, given the same level of resources (time lags \times reservoir dimension). We have now included references to [Duan et al., 2023] and [Dong et al., 2020] in the discussion:

This motivates exploring modifications of the original architecture to achieve the same performance with less resources. For example, [Duan et al., 2023] used reservoirs combined

with time delays, and [Dong et al., 2020] used structured matrices to speed up updates of the reservoir.

4. **The experiments conducted in this paper are primarily on relatively simple toy models, which do not adequately demonstrate the effectiveness and broad applicability of the method. Can the predictive capabilities of this method be showcased on high-dimensional systems (such as the Kuramoto-Sivashinsky (KS) chaotic system) and systems with observational noise?**

The major goals of the manuscript were to introduce our theoretical approach that explicitly connects next generation reservoir computing with the polynomial kernel machine, and to explain how to efficiently implement these networks on neuromorphic hardware. We primarily evaluated our approach on the same experiments from that study, and used simple models to demonstrate proof-of-principle. Still, your point is well taken, and we recognize the importance of testing the method on higher-dimensional dynamical systems. Following your suggestion, we have conducted additional experiments on the Kuramoto-Sivashinsky chaotic system for both our approach and alternative baselines. These experiments are reported in Section S-I in the revised Supplementary Material. The experiments demonstrate the importance of nonlinear features besides the lagged observations (Figure S.2) both for the conventional reservoir computing and for the investigated approaches with higher-order features. For the conventional reservoir computing, including $\tanh(\cdot)$ activation function was essential. Once included, it was possible to find a set of hyperparameters that performs well on the task, Figure S.1a. Similarly, the approaches with higher-order features perform well, Figure S.1b-c, once the second- & third- order features are considered in addition to the lagged observations. All three approaches performed equally well on this task but the conventional reservoir computing and the distributed representation approaches required about 40% fewer dimensions to achieve the performance matching to that of the product representation approach. These points are expressed in the new Section S-I:

The product and distributed representation approaches provide such nonlinear features as products between the time-delayed states. To achieve good performance with these approaches we had to form the feature space that includes first-, second-, and third- order features, $\mathcal{T} = (0, 1, 2)$, including only first-, and second- order features is not sufficient (not shown). The considered configuration amounts to $D = 6,545$ features for the product representation (red solid line) and provides much better predictive performance than the linear models. When the same feature space is instantiated with the distributed representation (green solid line), approximately equal performance is achieved already at $D = 4,000$ (i.e., about 40% fewer dimensions). For the echo state network approach, the nonlinear features are formed inside the reservoir through applying $\tanh(\cdot)$ to the random projections of the observable states. Similar to the product representation, using the echo state network with nonlinear features (black solid line) significantly improves upon its linear counterpart. While it performed slightly better than other approaches for the smallest dimensions tested, both the distributed representation and the echo state network approach achieve the predictive performance of the product representation at about $D = 4,000$ without significant improvements for larger values of D . Thus, for this task both implicit nonlinear features obtained via $\tanh(\cdot)$ activation and explicit polynomial features are equally amenable for obtaining strong predictive performance.

5. **Actually, the authors employ the delay embedding (see Eq. (4)), and the authors may discuss the difference between the proposed framework and Taken’s Embedding theory.**

Taken’s embedding theorem is a fundamental result which gives conditions under which the behavior of a dynamical system can be reconstructed from its lagged (delayed) observations, and provides an important theoretical basis for the use of lag/delay embeddings in reservoir computing [Takens, 1980]. The approach pursued in our manuscript includes both lags and products between them, an idea which was previously introduced in the context of reservoir computing in [Pyle et al., 2021, Gauthier et al., 2021] and found to be practically effective. Our additional experiments, which we ran to address items (4) and (6) (e.g., Sections S-I and S-V in the Supplementary Material), affirm this finding: models which include interactions between the lags perform better on the tasks we consider than the ones which only include lags. We do not claim this to be universally true, but our results, combined with the prior work in the area, suggest that the finding is robust enough to warrant inclusion in the empirical tool-kit of reservoir computing.

We, thus, accept the claim of [Gauthier et al., 2021] that products of lagged features are empirically useful. We seek to address a significant practical limitation in their use: the obvious way of implementing polynomial features by explicitly materializing them is not scalable when the number of lags used is large. Our contribution is to provide a more space-and-time efficient way to realize polynomial features using randomized representations which leads naturally to implementation in neuromorphic hardware. We believe our work significantly advances the understanding of design choices that are possible with reservoir networks and makes a practically useful contribution to the implementation thereof in novel kinds of hardware.

6. **Since the authors did not provide the detailed experimental configurations (a major concern), it is difficult to validate the correctness and generality of the results, and thus the value of this work is limited.**

This information was in Section 4.5, but the section title was probably misleading. Thus, we have now renamed this section to “Tasks and experimental configurations,” and we added further details about the choice of hyperparameters. We have also provided executable code of the experiments described in Figures 3, 4, 5, S.1, S.2, and S.3.

Additionally, this work lacks ablation studies on the influences of operations such as superposition, binding, and permutation. The robustness (e.g., random seed, noise) and hyperparameters (e.g., order) should be sufficiently evaluated.

Some of the suggested robustness studies were already present in the original manuscript. For example, the results in Figure 4 were computed from 1,000 random seeds. The effects of noise and amount of training data were considered in Sections S-V and S-VI in the Supplementary Material. In addition to these previously presented experiments, we conducted new experiments investigating i) the influence of polynomial order of features included in the feature space (Section S-V, Supplementary Material), ii) the relative performance of different models of vector symbolic architectures (Section S-VI, Supplementary Material), and iii) the influence of recursively permuting a random projection matrix while representing the memory buffer of the time-delayed states (Section S-VII, Supplementary Material).

7. **Though the authors introduce circular convolution, I wonder if this operator is necessary in the framework from both theoretical and experimental aspects.**

The binding operation is necessary theoretically to form distributed representations of polynomial features and to do well in practice (e.g., Figure S.4). Circular convolution is a specific way of implementing binding (but there are other ways, Sections 2.4 and S-VI). Regardless of the implementation, binding distributed representations has the same similarity structure as the explicit formation of the higher-order features. We use the block-wise circular convolution because of its sparsity and scaling properties that make it practical for neuromorphic hardware (Loihi 2).

8. **There is no evidence to support the claim “... is to protect representation of different time steps in the joint representation of the trajectory”.**

*This may be confusing as “protect” is the specialized term we use to describe different types of superposition properties in vector symbolic architectures’ distributed representations. Support for this claim comes from [Frady et al., 2023], which we now reference explicitly. Further support comes from some of our theoretical results: The first half of Section 4.4 demonstrates that the inner product between two input vectors (e.g., time-delayed states from the memory buffer) is preserved by their corresponding distributed representations. **Remark 1** in the second half of the section points out that permutations are a simple way to implement the “protected sum” representation. Furthermore, Section S-VII in the revised Supplementary Material demonstrates this experimentally. In order to make these points clearer, we added the references to both sources to the revised text:*

... is to protect representation of different time steps in the joint representation of the trajectory [Frady et al., 2023]. Thus, Eq. (17) is the randomized distributed representation corresponding to concatenation of several time points in the product representation, cf. Eq. (5). A theoretical argument supporting this claim is presented in the next section. The corresponding empirical evaluation is reported in Supplementary Material S-VII.

9. **Some symbols in the paper are reused, causing confusion. For example, the ‘i’ in Eq.(5, 7, 10, 11, ...) represents the time index of G, while also denotes the elements in the set M. The bold font of the first term in the equation after the sentence ‘... is defined recursively as’ is not correct.**

Thank you for noticing these issues. We have performed an additional check in order to find and resolve these instances of potentially confusing notation.

Reviewer 2

In their manuscript “Principled Neuromorphic Reservoir Computing,” Kleyko and co-authors propose what might be considered a third-generation form of reservoir computing, combining the randomized distributed representations of traditional reservoir computing (RC) with the higher-order input features used to great effect in by a nonlinear vector autoregression (NVAR) machine (termed “NG-RC” in Ref. 11 and “product representation” in the current work.). In this way, the authors’ “distributed representation” reservoir attained comparable or better NMSEs on classic RC benchmarks using up to or fewer input features compared with the product representation method. The dimension of the input feature space was changed to be a tunable parameter by recognizing that distributed representations incorporating vector symbolic architectures (VSA) operations approximate the same similarity structure of product representations, notably through the superposition (or collective state computing) properties of VSA. Further, the authors propose and demonstrate a neural network implementation of this distributed representation reservoir of Sigma-Pi neurons on Intel’s Loihi2 hardware, whose simulation code is available on github (though not the LAVA-VSA toolbox). This demonstration highlighted the power of VSA operators for preserving dimensionality unlike tensor products, such that neural subnets can be trivially reused.

Figure 1 does an excellent job illustrating the key differences and similarities among traditional RC, product representation, and distributed representation approaches. Unlike the NG-RC, which could be argued as strictly better than traditional RC for prediction in terms of parameter tuning and training data efficiency, the distributed representation did not uniformly perform better than the NG-RC (product representation), though it was at minimum comparable. A critical contribution of this paper then is also to unify the mathematical descriptions of each approach, thereby, providing a more formal explanation for the comparable behavior via different means. As the authors’ note, the product representation’s feature size was often a good initial estimate for the distributed representation.

The manuscript is very well written, though dense and detailed, as necessary to clearly detail the mathematical underpinnings. I particularly appreciate the expanded equations in section 4 to ensure the reader could follow the terms intended by the notation. Due to the popularity of RC and the advantages of NG-RC (product representation), this distributed representation and mathematical interpretation will be of great interest to the RC community and readers of Nature Communications. Aside from minor questions below, I recommend publication.

We highly appreciate the precise, exciting, and for us highly encouraging summary of the manuscript. We can only hope that this manuscript can live up to this summary and will enrich the reservoir computing community and readers of Nature Communications both practically and conceptually. Please find below our notes on how we implemented your suggestions. The modified text has been colored in blue.

1. As far as I understand, the results from Fig. 3 were not computed on the Intel Loihi2 hardware, only Fig. 5 results. Section 2.6 is described as an ablation study, but since the authors’ highlight the size of the Intel Hala Point, it is noticeable that they chose to showcase this large-scale hardware on an ostensibly simpler problem (Fig. 5). It would seem the neuromorphic hardware would be large enough to perform the chaotic time series prediction tasks (Fig. 3). Did the authors try to replicate the results on Fig. 3 as well on the Hala Point?

As noted by the reviewer, our intention with Section 2.6 was to demonstrate the feasibility of implementing the proposed networks of Sigma-Pi neurons on neuromorphic hardware. The intention with highlighting the size of the Intel Hala Point was to demonstrate that such hardware is getting more mature and provides substantial scalability. But this does cause dissonance between the complexity of the task and the scale of hardware in our demonstrations. We adjusted the formulations in the caption of Fig. 5 to reduce this dissonance. To answer your question about performing the chaotic time series prediction tasks on neuromorphic hardware, we are confident that Loihi 2 is large enough for doing so – thousands of chips can be scaled into a large system as exemplified by Hala point. The primary reason for not replicating the results of Fig. 3 is of the engineering nature, as significant efforts are still needed to build the full-fledged system. This implementation would not add substantially to the key messages of the manuscript, so we have traded off by demonstrating the key step of the system (i.e., evolution of the reservoir dynamics) at smaller scale.

2. **The NG-RC paper (Ref. 11) argued that for traditional RC, multiple reservoirs had to be created and parameters tuned to find an optimal RC solution. Their proposed higher-order input thus skipped the variability of the input and reservoir weights entirely, simplifying training. Yet, in this paper, the authors claim that the distributed representation sufficiently approximates the kernel regardless of the random vectors used, thereby, requiring minimal tuning parameters, i.e. the input feature size D and regularization parameter α . It may be worth a remark about the optimization search spaces between the two methods.**

Indeed, when using the distributed representations, the only additional parameter is the dimensionality D (since α is also used in NG-RC). We suggest that it is not hard to tune it, since the dimensionality of the corresponding NG-RC reservoir (follows the combinatorics of polynomial features) provides an initial conservative estimate of the required dimension. The last paragraph of the discussion attempted to make this point, but no proper contrast was made to NG-RC. This part was, therefore, expanded in the revised text:

These results emphasize the role of dimensionality as a tuneable hyperparameter of the proposed approach. Note that this is the only additional hyperparameter introduced beyond the hyperparameters in the product representation scheme (i.e., the choice of delayed states, order of polynomial features, and the regularization parameter) [Gauthier et al., 2021]. As follows from the results in Figure 4, the dimensionality of randomized representations does not require extensive tuning. A simple heuristic is to initially use the number of features in the product representation, a conservative estimate that can often be reduced in practice. Thus, the proposed approach introduces minimal overhead to the hyperparameter search space compared to that of the product representation scheme.

Typos: Section 2.3 “and, in essence, form”

Thank you for spotting this typo. It has been corrected.

Suggestions:

1. **The Fig. 5 caption or legend should include a remark about the dashed line being CPU results.**
We have revised the caption for Fig. 5c accordingly.
2. **The Fig. S.2 caption or legend should include a remark about the grey dash-dot line being structured results. In the spirit of random connections with reservoir computing, I found this insight particularly interesting.**
The caption for Fig. S.2 have been revised accordingly.

References

- [Barbosa and Gauthier, 2022] Barbosa, W. A. S. and Gauthier, D. J. (2022). Learning spatiotemporal chaos using next-generation reservoir computing. *Chaos: An Interdisciplinary Journal of Nonlinear Science*, 32(9):1–11.
- [Dong et al., 2020] Dong, J., Ohana, R., Rafayelyan, M., and Krzakala, F. (2020). Reservoir computing meets recurrent kernels and structured transforms. In *Advances in Neural Information Processing Systems (NeurIPS)*, pages 16785–16796.
- [Duan et al., 2023] Duan, X.-Y., Ying, X., Leng, S.-Y., Kurths, J., Lin, W., and Ma, H.-F. (2023). Embedding theory of reservoir computing and reducing reservoir network using time delays. *Physical Review Research*, 5(2).
- [Frady et al., 2023] Frady, E. P., Kleyko, D., and Sommer, F. T. (2023). Variable binding for sparse distributed representations: Theory and applications. *IEEE Transactions on Neural Networks and Learning Systems*, 34(5):2191–2204.
- [Franz and Schölkopf, 2006] Franz, M. O. and Schölkopf, B. (2006). A unifying view of Wiener and Volterra theory and polynomial kernel regression. *Neural Computation*, 18(12):3097–3118.
- [Gauthier et al., 2021] Gauthier, D. J., Bollt, E., Griffith, A., and Barbosa, W. A. S. (2021). Next generation reservoir computing. *Nature Communications*, 12(1):1–8.
- [Pyle et al., 2021] Pyle, R., Jovanovic, N., Subramanian, D., Palem, K. V., and Patel, A. B. (2021). Domain-driven models yield better predictions at lower cost than reservoir computers in Lorenz systems. *Philosophical Transactions of the Royal Society A*, 379:1–22.
- [Takens, 1980] Takens, F. (1980). Detecting strange attractors in turbulence. In *Dynamical Systems and Turbulence*, pages 366–381.